# Interleukin-18 produced by bone marrow-derived stromal cells supports T-cell acute leukaemia progression

Benjamin Uzan[1,2,3,4], Sandrine Poglio[1,2,3,4], Bastien Gerby[1,2,3,4], Ching-Lien Wu[1,2,3,4], Julia Gross[1,2,3,4], Florence Armstrong[1,2,3,4], Julien Calvo[1,2,3,4], Xavier Cahu[1,2,3,4], Caroline Deswarte[5], Florent Dumont[6,7,8], Diana Passaro[9,10,11], Corinne Besnard-Guérin[6,7,8], Thierry Leblanc[12], André Baruchel[12], Judith Landman-Parker[5], Paola Ballerini[1,2,3,4,5], Véronique Baud[6,7,8], Jacques Ghysdael[9,10,11], Frédéric Baleydier[13,†], Francoise Porteu[6,7,8,†] & Francoise Pflumio[1,2,3,4,*]

## Abstract

Development of novel therapies is critical for T-cell acute leukaemia (T-ALL). Here, we investigated the effect of inhibiting the MAPK/MEK/ERK pathway on T-ALL cell growth. Unexpectedly, MEK inhibitors (MEKi) enhanced growth of 70% of human T-ALL cell samples cultured on stromal cells independently of NOTCH activation and maintained their ability to propagate *in vivo*. Similar results were obtained when T-ALL cells were cultured with *ERK1/2*-knockdown stromal cells or with conditioned medium from MEKi-treated stromal cells. Microarray analysis identified interleukin 18 (IL-18) as transcriptionally up-regulated in MEKi-treated MS5 cells. Recombinant IL-18 promoted T-ALL growth *in vitro*, whereas the loss of function of IL-18 receptor in T-ALL blast cells decreased blast proliferation *in vitro* and in NSG mice. The NFKB pathway that is downstream to IL-18R was activated by IL-18 in blast cells. IL-18 circulating levels were increased in T-ALL-xenografted mice and also in T-ALL patients in comparison with controls. This study uncovers a novel role of the pro-inflammatory cytokine IL-18 and outlines the microenvironment involvement in human T-ALL development.

**Keywords** IL-18; inflammation; stromal cells; T-ALL

**Subject Categories** Cancer; Haematology

## Introduction

T-cell acute lymphoblastic leukaemia (T-ALL) is an aggressive haematologic malignancy accounting for about 15% of paediatric and 25% of adult ALL (Pui *et al*, 2004). T-ALL originates from a block of differentiation and uncontrolled proliferation of developing T cells and is lethal without therapy. Current chemotherapies provide an overall survival (OS) higher than 75% in children (Pui *et al*, 2008) and of about 50% in adults at 5 years (Marks *et al*, 2009). Both paediatric and adult relapses have a very poor outcome. Therefore, the identification of molecular targets and the development of specific therapies are major goals that require a better understanding of T-ALL molecular mechanisms.

In T-ALL, many oncogenic pathways are often activated due to chromosomal abnormalities (Ferrando *et al*, 2002), and the disease can thus be classified into distinct molecular groups with specific gene expression signatures (Van Vlierberghe & Ferrando, 2012).

1  Commissariat à l'Energie Atomique et aux Energies Alternatives (CEA), DSV-IRCM-SCSR-LSHL, Equipe Labellisée Ligue Contre le Cancer, UMR 967, Fontenay-aux-Roses, France
2  INSERM, U967, Fontenay-aux-Roses, France
3  Université Paris Diderot, Sorbonne Paris Cité, UMR 967, Fontenay-aux-Roses, France
4  Université Paris-Sud, UMR 967, Fontenay-aux-Roses, France
5  Service D'hématologie Pédiatrique, Assistance Publique – Hôpitaux de Paris, Hôpital A. Trousseau, Paris, France
6  INSERM U1016, Institut Cochin, Paris, France
7  CNRS UMR8104, Paris, France
8  Université Paris Descartes, Sorbonne Paris Cité, Paris, France
9  Institut Curie, Centre Universitaire, Orsay, France
10 CNRS UMR 3306, Orsay, France
11 Institut National de la Santé et de la Recherche Médicale U1005, Orsay, France
12 Service D'hématologie Pédiatrique, Assistance Publique – Hôpitaux de Paris, Hôpital Robert Debré, Paris, France
13 Institut d'Hématologie et Oncologie Pédiatrique, Hospices Civils de Lyon et Université Claude Bernard, Lyon, France
   *Corresponding author. Tel: +33 1 46 54 86 17; Fax: +33 1 46 54 91 38; E-mail: francoise.pflumio@cea.fr
   †Co-authorship.

Transcription factors, among which TAL1 (T-cell acute lymphocytic leukaemia 1; also known as SCL) and HOX11L2 (homeobox-11/L2; also known as TLX3), are frequently deregulated (Graux *et al*, 2006). Signalling pathways also are often perturbed. For instance, *PTEN* mutations are common in leukaemic blasts and may lead to the activation of the PI3K/AKT pathway (Palomero *et al*, 2007; Silva *et al*, 2008).

The mitogen-activated protein kinase (MAPK) signalling cascade, one of the pathways that mediate proliferation and differentiation of hematopoietic cells (Lee & McCubrey, 2002), is frequently deregulated in adult T-ALL. Among the MAPK pathways, the extracellular signal-regulated kinases 1 and 2 (ERK1/2), two serine/threonine protein kinases, the activation of which depends on the upstream MEK1/2 tyrosine/threonine kinases, are constitutively phosphorylated in approximately 38% of patients with T-ALL (Gregorj *et al*, 2007). Moreover, functional activation of some oncogenic proteins, such as TAL1/SCL, requires phosphorylation by MEK/ERK kinases (Cheng *et al*, 1993; Wadman *et al*, 1994; Talora *et al*, 2006). MAPK inhibitors are thus interesting molecules for T-ALL treatment.

Besides the well-accepted implication of intracellular oncogenic pathways in T-ALL development, there is increasing evidence that extracellular signals provided by the microenvironment also participate in T-ALL onset/progression (Nwabo Kamdje & Krampera, 2011). Signals initiated by growth factors, such as interleukin 7 (IL-7; Barata *et al*, 2001) or insulin growth factor 1 (IGF-1; Medyouf *et al*, 2011), or by receptor–ligand interactions, such as NOTCH-Delta-like1/4 (DL1/4; Weng *et al*, 2003; Armstrong *et al*, 2009), are crucial for T-ALL development. The identification of mutations in the genes encoding components of these signalling pathways further confirms their importance in T-ALL (Weng *et al*, 2004; Shochat *et al*, 2011; Zenatti *et al*, 2011). Biological phenomena, such as inflammation, can also participate in cancer progression, particularly in the case of haematological malignancies, such as chronic myelomonocytic leukaemia and lymphomas (Alexandrakis *et al*, 2004; Liu *et al*, 2011; Reynaud *et al*, 2011). However, it is not known whether an inflammatory environment plays a role in T-ALL development/progression.

In the present work, we investigated whether inhibition of the MEK/ERK pathway could hinder T-ALL progression. Unexpectedly, MEK inhibition (MEKi) increased T-ALL cell proliferation. We then demonstrate that this effect is due to MEKi-mediated increased secretion of the pro-inflammatory cytokine IL-18 by stromal cells. Moreover, patients with T-ALL and immunodeficient mice xenografted with human T-ALL cells have higher circulating levels of IL-18 than controls. Although still preliminary, comparison of disease-free survival in T-ALL patients stratified according to the European Group of Immunological Classification of Leukaemia (EGIL) relative to their high/low IL-18 plasmatic level suggests that IL-18 levels may be of prognosis value in T-ALL. This work demonstrates the importance of the pro-inflammatory IL-18 cytokine and the major role of the microenvironment in human T-ALL development/progression.

## Results

### MEK inhibition promotes human T-ALL cell growth *in vitro* and *in vivo*

To test the effect of MEK inhibition in T-ALL, human T-ALL cell samples were co-cultured with MS5 or MS5-DL1 stromal cells

directly after isolation from patients or after injection in immuno-deficient mice (see Supplementary Table S2 for a description of the 26 T-ALL samples used in this study) in the presence of a MEKi (U0126, PD98059 or PD184352). T-ALL cells were then counted by FACS analysis (CD45$^+$CD7$^+$ cells) or trypan blue staining at different time points. Overall 20/26 (77%) T-ALL cell samples could be cultured *in vitro* and 69% (11/16) of them proliferated better when grown on MS5-DL1 than on MS5 cells (Table 1). Surprisingly, cell growth was increased in 14/20 (70%) of MEKi-treated T-ALL samples compared to untreated co-cultures (Table 1 and Fig 1A). Similar results were obtained with the three MEKi PD184352, PD98059 and U0126 (Supplementary Fig S1 and not shown) and with two other cell feeders (mouse stromal OP9 cells and human mesenchymal stem cells; Supplementary Fig S2). As MEKi effect on proliferation was milder in T-ALL cells

**Table 1. Growth response of T-cell acute lymphoblastic leukaemia (T-ALL) cells to MEK inhibition**

| T-ALL samples | Co-culture with | | | |
| | MS5 cells | | MS5-DL1 cells | |
| | Diluent | MEKi | Diluent | MEKi |
|---|---|---|---|---|
| M18 | 0 | +++ | + | ++ |
| M18 x | ++ | ++++ | +++ | ++++ |
| M22 | 0 | ++ | 0 | ++ |
| M30 | + | ++ | ++ | +++ |
| M34 | 0 | + | +++ | +++ |
| M40 | 0 | + | + | ++ |
| M69 x | ++ | +++ | +++ | +++ |
| M78 x | ++ | +++ | +++ | +++ |
| M86 | 0 | + | 0 | + |
| M97 | + | ++ | +++ | ++++ |
| M105 | +++++ | +++++ | +++++ | +++++ |
| M105 x | ++++ | ++++ | +++ | nd |
| M106 x | +++ | ++++ | ++++ | ++++ |
| M108 | ++++ | +++++ | nd | nd |
| M109 | 0 | 0 | + | ++ |
| M110 | 0 | + | nd | nd |
| M112 x | + | + | nd | nd |
| M114 | +++ | ++ | +++ | nd |
| M118 | 0 | + | +++ | nd |
| M120 | + | ++ | nd | nd |
| M121 | 0 | 0 | + | + |
| M129 x | +++ | +++ | +++ | nd |

Human T-ALL cells isolated from newly diagnosed patients or from xenografted mice (x) were seeded (2 × 10$^5$/well in 24-well plates) on MS5 or MS5-DL1 stromal cells and cultured in the presence of MEKi (1 μM PD184352) or diluent alone (DMSO, negative control) for 28 days. CD45+CD7+ leukaemic cells were counted at day 28 and (+), (++), (+++), (++++), (+++++) indicate, respectively, a < 10-fold, between 10- and 50-fold, 50- and 250-fold, 250- and 1,000-fold and higher than 1,000-fold number increase compared to the original number of seeded cells. 0, no cells were recovered after culture; nd, not done.

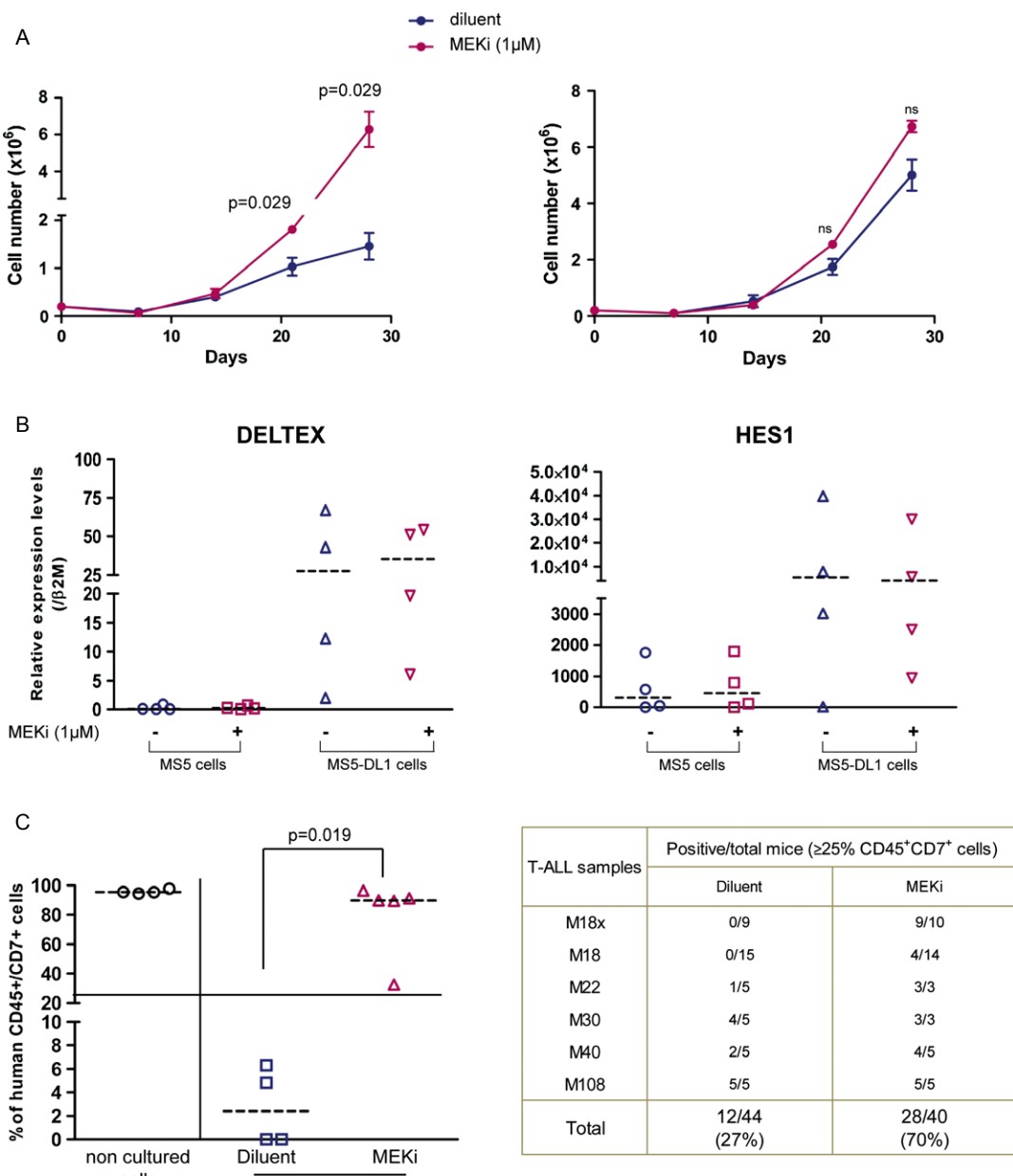

**Figure 1. MEK inhibition stimulates the proliferation of T-ALL cells co-cultured with MS5 stromal cells and maintains T-LIC activity.**

A T-ALL cells were cultured on MS5 (left panel) or MS5-DL1 (right panel) stromal cells in the presence (MEKi) or not (DMSO alone) of 1 μM PD184352 (MEK inhibitor) for 28 days. Cells were counted by harvesting individual wells from 24-well/plates each week and using trypan blue or FACS. Shown are the cumulative data (mean ± s.d. of triplicates) for the M69 T-ALL sample as representative of the 14 MEKi-responsive T-ALL samples.

B *HES1* and *DELTEX* expression levels in four individual T-ALL samples after 28 days of culture with MS5 or MS5-DL1 cells in the presence or absence of PD98059 (MEKi).

C Engraftment level in bone marrow of T-ALL cells that were cultured with or without MEKi (PD98059: M18, M22, M40; PD184352: M18, M108; U0126: M18, M30) for 1–2 weeks before injection. Left panel shows a representative experiment in which M18x T-ALL cells were injected in NSG mice directly or after 2 weeks of co-culture with MS5 cells in the presence or not of PD98059. The right panel provides a summary of the engraftment level of all tested T-ALL samples. Transplantations of M18, M22 and M30 T-ALL treated with PD98059 and U0126 were done in NS mice (Mann–Whitney non-parametric test).

cultured on MS5-DL1 than on MS5 cells (Fig 1A and Table 1), we asked whether NOTCH was implicated in MEKi effect. MEKi did not significantly modify the expression of the NOTCH target genes *DELTEX*, *HES1* and *PTalpha* during culture (Fig 1B and not shown), indicating that NOTCH is not involved in MEKi effect. In addition, MEKi-mediated increase in T-ALL cell proliferation did not correlate with specific *NOTCH1* mutations or oncogene activation (Supplementary Table S2).

Then, MEKi-treated or untreated T-ALL cells (*n* = 6 different samples) were injected in immunodeficient NSG mice and leukaemia development was monitored after 6–8 weeks. Four of the six MEKi-treated T-ALL samples showed a more aggressive behaviour than

untreated cells. Overall, significant bone marrow invasion by leukaemic blasts ($\geq 25\%$ CD45$^+$CD7$^+$ cells) was observed in 70% (28/40) of mice transplanted with MEKi-treated T-ALL cells compared to 27% (12/44) of animals injected with untreated cells (Fig 1C).

## Treatment of stromal cells with MEK inhibitors increases T-ALL cell growth and enhances IL-18 production

As ERK1/2 phosphorylation was decreased in T-ALL blasts but also in MS5 cells following incubation with the MEKi PD184352 (Fig 2A), we investigated whether MEKi effect on T-ALL proliferation was mediated through MS5 cells. To test this hypothesis, conditioned medium (CM), which was harvested every other day from MS5 cultures grown in the presence of MEKi, was added to untreated co-cultures of T-ALL and MS5 cells. Three independent experiments showed that the addition of CM from MEKi-treated MS5 cells was sufficient to increase T-ALL cell proliferation (Fig 2B and data not shown). Similarly, co-culture of T-ALL samples with MS5 cells in which *ERK1* and/or *ERK2* were silenced (shERK1/2 MS5 cells) resulted in a significant increase in blast cell proliferation compared to control co-cultures (shCTL MS5; Fig 2C and Supplementary Fig S3). This effect was particularly strong upon *ERK2* silencing. However, it should be noted that shERK1/2 MS5 cells were less viable (not shown), thus explaining their relative lower support of T-ALL growth compared to shERK2 MS5. Altogether, these results show that inhibition of the MEK/ERK MAPK pathway induces the secretion by MS5 feeder cells of factors that promote T-ALL proliferation.

To identify these factors, we performed gene expression profiling of MS5 cells treated or not with MEKi for 7 days (three independent experiments) and identified 110 genes, the expression of which was modified upon incubation with MEKi (Supplementary Fig S4 and Fig 2D). Among the up-regulated genes that encode secreted factors, we selected the pro-inflammatory cytokine IL-18 as it participates in tumour growth and metastasis formation (Kim *et al*, 2006, 2007; Amin *et al*, 2007). In accordance with the microarray results, a mean 5.8-fold increase (range: 2.65–10.7) of IL-18 gene expression was observed in MS5 cells upon MEKi treatment (Fig 2E). Similarly, IL-18 protein concentration was 2.3- and 2.1-fold higher in CM from PD184352-treated MS5 and from shERK1/2 MS5 cells, respectively, than in CM from control MS5 cells (Fig 2F).

## Stromal cell-secreted IL-18 contributes to human T-ALL growth *in vitro*

To determine whether MS5-secreted IL-18 could affect T-ALL cell proliferation, the expression of the α and β chains of the IL-18 heterodimer receptor (IL-18Rα and IL-18Rβ) was quantified in T-ALL samples and compared with that of normal umbilical cord blood (UCB) CD7+ T cells (Fig 3A and not shown). The expression differences probably reflect differences in maturation stages of the two cell types as IL-18 promotes mature T-helper cell generation (Nakanishi *et al*, 2001).

Then, T-ALL/MS5 co-cultures were grown in the presence or not (control) of recombinant human IL-18 for 4 weeks. Cell proliferation was increased in six of 10 tested T-ALL samples, compared to controls (Fig 3B and C and data not shown). Moreover, IFN-γ expression, a downstream target of the IL-18 pathway (Okamura *et al*, 1995), was significantly higher in T-ALL cells cultured in the presence of IL-18 than in controls, indicating that the IL-18 pathway is activated in T-ALL cells in response to exogenous IL-18 (Fig 3B and data not shown).

The role of IL-18 in T-ALL cell proliferation was further confirmed by the finding that addition of an anti-mouse IL-18 neutralizing antibody (mIL-18Ab) decreased MEKi-induced T-ALL cell proliferation in T-ALL/MS5 co-cultures (Fig 3C and Supplementary Fig S5). Similar results were obtained in the five tested MEKi-responding T-ALL samples. Moreover, mIL-18Ab could reduce T-ALL cell proliferation in comparison with control cells even in the absence of MEKi (Fig 3C and Supplementary Fig S5). This effect was variable, but was particularly strong in the M105 T-ALL sample (Supplementary Fig S5B), which shows high T-ALL initiating cell (T-LIC) activity (Gerby *et al*, 2011). In agreement with IL-18 inhibition, IFN-γ levels were lower in blast cells treated with mIL-18Ab compared to controls (Supplementary Fig S5C). Finally, the role of stromal cell-derived IL-18 in T-ALL cell proliferation was also explored in co-cultures of T-ALL and MS5 cells in which *IL-18* was silenced (shIL-18). A 75% decrease in IL-18 expression in MS5 cells significantly reduced T-ALL cell proliferation on its own and also strongly reduced MEKi proliferative effect in comparison with co-cultures with shCTL MS5 cells (Fig 3D). Similar results were obtained also when intracellular Notch1 (ICN1)-induced T-ALL mouse cells were co-cultured with shCTL or shIL-18 MS5 cells (Supplementary Fig S6).

---

**Figure 2.  MEKi induces IL-18 production by MS5 stromal cells.**

A   Western blot analysis of protein lysates from M97 T-cell acute lymphoblastic leukaemia (T-ALL) cells co-cultured with MS5 or MS5-DL1 (left panel) and from MS5 or MS5-DL1 cells (right panel) incubated (+) or not (−) with the MEKi PD184352 for 1 h before harvesting.

B   Assessment of M18 T-ALL cell proliferation when cultured in the presence of conditioned medium (CM) from control (DMSO) (CM CTL) or MEKi-treated (1 μM PD184352, CM MEKi) MS5 cells. Data are representative of three experiments performed in triplicate. Inset shows the details of day 21 (mean ± s.d.; Mann–Whitney non-parametric test).

C   Proliferation of M69 T-ALL cells co-cultured with shERK1- and/or shERK2-transduced MS5 cells (inset shows the reduction of ERK1/2 protein expression following transduction with the different short hairpin RNAs). Data are representative of three experiments performed in triplicate (mean ± s.d.; Mann–Whitney non-parametric test). tERK1/2, total ERK1/2 proteins.

D   Microarray analysis (Affymetrix arrays) of MS5 cells cultured in the presence of 1 μM PD184352 (MEKi) or diluent for 7 days. Heatmap shows the transcripts, the expression of which was most significantly different in the two conditions (arrow indicates IL-18).

E   *IL-18* mRNA expression in MS5 cells grown in the presence of 1 μM PD184352 or not (diluent) for 1 week (mean ± s.d. of five independent experiments).

F   IL-18 protein expression in the supernatants of MS5 cells treated with 1 μM PD184352 or diluent for 1 week and in the supernatants of MS5 cells 1 week after transduction with shERK1+2 or shCTL (mean ± s.d. of triplicates; Mann–Whitney non-parametric test).

Source data are available for this figure.

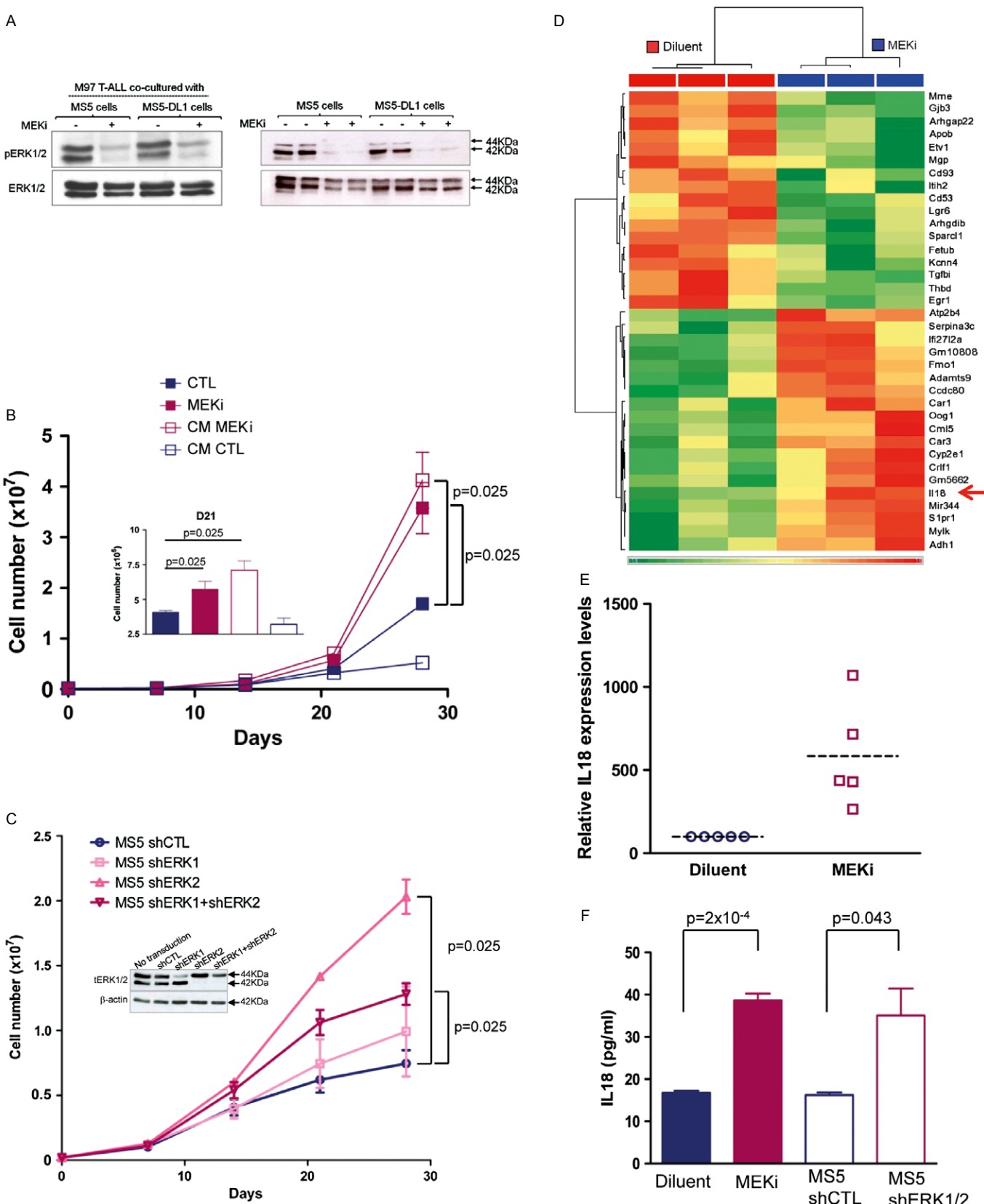

**Figure 2.**

As abnormal constitutive NF-κB activation plays an important role in controlling T-ALL cell proliferation (Kordes *et al*, 2000), we hypothesized that IL-18 may activate NF-κB. Indeed, recombinant IL-18 led to a significant increase in NF-κB DNA binding activity in T-ALL cells (Supplementary Fig S5D and E). Moreover, both basal and IL-18-induced NF-κB activation were higher in M105 T-ALL cells co-cultured with MS5 cells than without. Importantly, NF-κB DNA binding activity was strongly reduced when T-ALL cells were cultured with shIL-18 MS5 cells in comparison with shCTL MS5 cells

(Supplementary Fig S5E). NF-κB activation in T-ALL was also observed when blast cells were cultured with CM from MEKi-treated MS5 cells, supporting a link between MEKi, IL-18 and NF-κB activation (Supplementary Fig S5F).

IL-18Rα silencing in T-ALL cells further supported the involvement of stromal IL-18 in leukaemic cell proliferation (Supplementary Fig S7). Indeed, proliferation of shIL-18Rα T-ALL cells was reduced compared to shCTL T-ALL cells (Fig 3E and F). Less than 2% of T-ALL cells were GFP negative (thus not infected) after

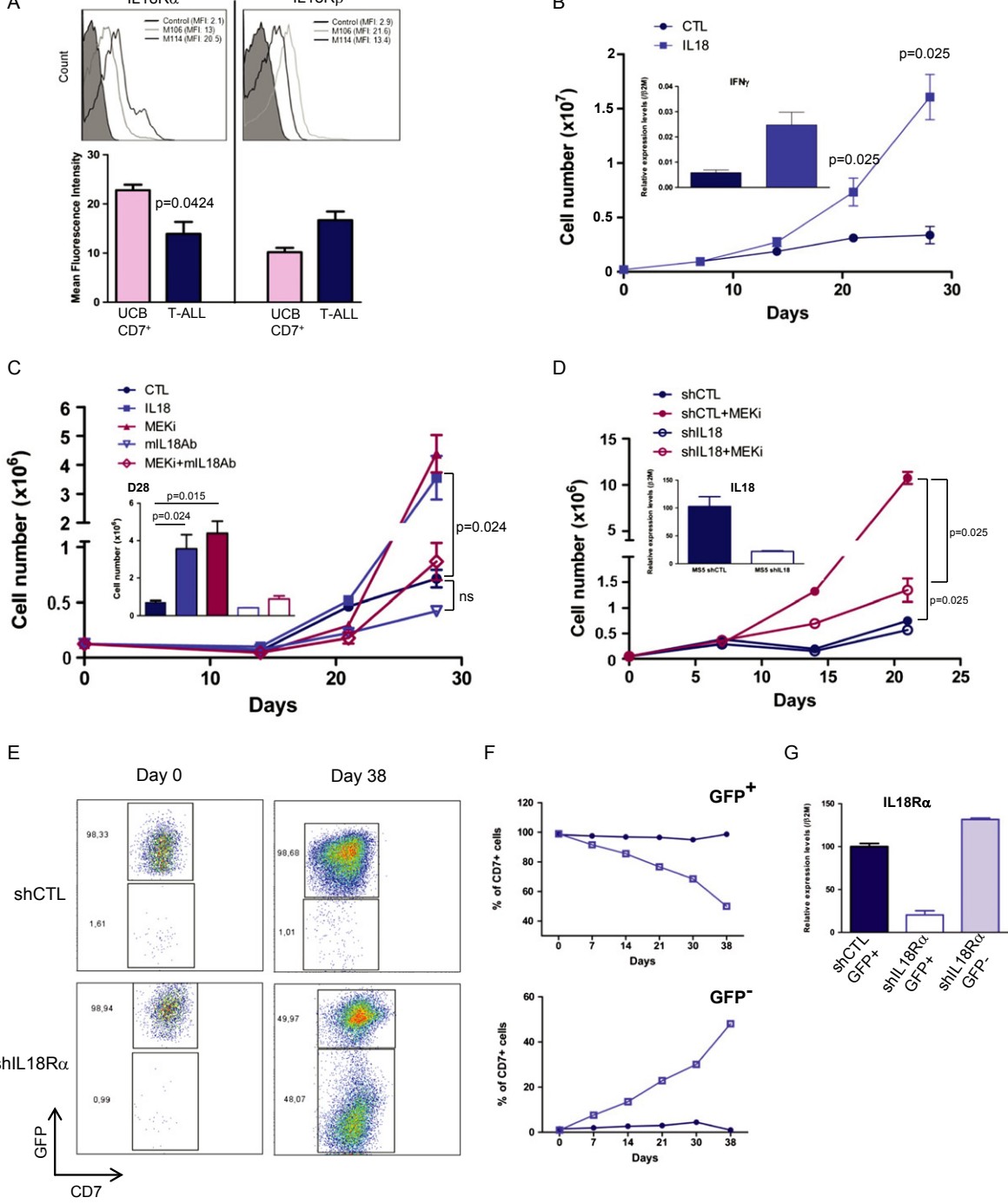

**Figure 3.**

transduction with shIL-18Rα or shCTL (Fig 3E, left panels) and this level progressively raised to 48% in T-ALL cells transduced with shIL-18Rα, while it remained stable (1.2%) in T-ALL cells transduced with shCTL at day 38 of co-culture with MS5 cells (Fig 3E, right panels and Fig 3F). At the end of the experiment, it was confirmed that IL-18Rα expression was strongly decreased only in shIL-18R/GFP+ T-ALL cells, but not in shCTL/GFP+ and shIL-18Rα/ GFP$^-$ T-ALL cells (Fig 3G). Altogether, these data indicate that stromal IL-18 contributes to human T-ALL cell proliferation, possibly through activation of the NF-κB pathway.

## Interfering with the IL-18 pathway in human T-ALL cells delays leukaemia development *in vivo*

We next investigated whether interfering with the IL-18 pathway affects leukaemia development *in vivo*. First, NSG mice were injected with three T-ALL cell samples pre-treated or not (control) with mIL-18Ab for 2 weeks. Bone marrow monitoring at week 6 post-xenograft indicated that the leukaemic load was significantly lower in two of the three pre-treated T-ALL samples in comparison with controls (Fig 4A). However, animals' survival was not improved (Fig 4B). This indicates that transient pre-treatment with mIL-18Ab *in vitro* does not durably interfere with T-ALL cell expansion *in vivo*. Then, NSG mice were transplanted with shIL-18Rα T-ALL or shCTL T-ALL (control) cells. In agreement with our *in vitro* results (Fig 3E and F), CD7+GFP$^-$ leukaemic blasts showed a growth advantage over CD7+GFP+ cells in several animals injected with shIL-18Rα T-ALL cells compared to control mice (Fig 4C). Moreover, the spleens from mice transplanted with shIL-18Rα T-ALL cells were significantly smaller than those of control animals (Fig 4D). Survival of mice injected with shIL-18Rα T-ALL cells was significantly improved compared to controls (Fig 4E). Altogether, these data indicate that IL-18 contributes to T-ALL progression in xenograft models.

## IL-18 levels are increased in T-ALL-xenografted mice and in patients with T-ALL

Finally, measurement of IL-18 levels in serum samples of xenografted mice indicated that IL-18 concentration was higher (threefold) in NSG mice injected with human T-ALL cells ($n = 20$; six different T-ALL samples) compared to age-matched non-transplanted mice

($n = 19$; Fig 5A), suggesting that T-ALL development drives IL-18 production in mice. Interestingly when looking into mouse IL-18 production either in tumour-free (Fig 5B) or in tumour-bearing (not shown) NSG mice, we found that haematopoietic CD45$^+$ and non-haematopoietic CD45$^-$ cells could produce IL-18, suggesting that BM cells constitute a reservoir of IL-18 (Arend *et al*, 2008). Similarly, IL-18 plasma levels were significantly increased also in patients with T-ALL (mean ± s.d.: 337 ± 34 pg/ml; $n = 92$) compared to healthy paediatric controls (mean ± s.d.: 71 ± 12 pg/ml; $n = 29$; Fig 5C). Moreover, IFN-γ plasma levels, an indication of IL-18 pathway activation (Okamura *et al*, 1995), were also significantly increased in patients with T-ALL ($n = 25$) compared to controls ($n = 15$; Fig 5D), and probably as a consequence also there was an increase in IL-18-binding protein (IL-18BP, Supplementary Fig S8; Dinarello *et al*, 2013). Interestingly, free IL-18 levels, calculated according to (Novick *et al*, 2001), showed an excellent correlation with total IL-18 concentrations (Supplementary Fig S8).

To determine the clinical impact of circulating IL-18 levels, the clinical features of 81 patients with T-ALL were analysed relative to their IL-18 plasma levels. Comparative analysis of patients with high (over the median value, $n = 40$ patients) and low (under the median value, $n = 41$ patients) IL-18 plasma levels found no correlation with age, sex, CNS involvement, white blood cell count, EGIL classification, cytogenetic abnormalities, and response to prephase treatment (Table 2). Mediastinal enlargement/effusions were more frequent in the low IL-18 than in the high IL-18 group (Table 2). Likewise, the low IL-18 group of patients included 5/6 T-lymphoblastic lymphomas. Disease-free survival was not impacted by the IL-18 secretion level (not shown). As it was reported that the immunophenotype could influence the outcome of patients with T-ALL (Niehues *et al*, 1999), we tested the prognostic impact of the EGIL TIII and not-TIII (i.e. T I/II/IV) phenotypes. While no significant difference in the whole population was observed (data not shown), when IL-18 levels are combined with EGIL stratification in a Cox-regression model, EGIL TIII IL-18 low patients tend to show a better prognosis than EGIL not-TIII low IL-18 patients [HR = 3.9, 95% IC (1.02–18), $P = 0.046$] and in a much lesser extent than high IL-18-EGIL TIII [HR = 2.5, 95% IC (0.55–11), $P = 0.24$] and high IL-18-EGIL not-TIII [HR = 2.5, 95% IC (0.63–10), $P = 0.19$; Supplementary Fig S9]. These data remain preliminary and further analyses remain to be performed before to conclude on a prognosis

**Figure 3. IL-18 promotes T-cell acute lymphoblastic leukaemia (T-ALL) growth *in vitro*.**

A    Cell surface expression of IL-18Rα (left panel) and IL-18Rβ (right panel) in M106 and M114 T-ALL cells. Histograms show the mean fluorescence intensity (MFI) of IL-18Rα and IL-18Rβ staining in, respectively, 7 and 11 T-ALL samples. Control: T-ALL or UCB CD7+ T cells stained with isotype control antibodies.

B    IL-18 stimulates T-ALL cell proliferation. $2.10^5$ M18 T-ALL cells were cultured on MS5 and in the presence or not (CTL) of recombinant human IL-18 for 28 days. Cells were harvested every week, counted and re-plated. Data are representative of five experiments performed in triplicate. Inset shows the quantification of *IFN*-γ mRNA expression levels in IL-18-treated and control T-ALL samples at day 28 of culture.

C    M30 T-ALL proliferative response when cultured in the presence of MEKi (1 μM PD184352), recombinant IL-18 or an antibody against mouse IL-18 (mIL-18Ab) for 28 days. Data are the mean ± s.d. of triplicates. Results are representative of 2–5 T-ALL samples/condition. n.s., not significant.

D    Proliferative response of M105 T-ALL cells cultured on MS5 cells in which *IL-18* was silenced (shIL-18) or not (shCTL) and in the presence or not of MEKi (1 μM PD184352) for 21 days. Inset shows the quantification of *IL-18* mRNA in MS5 cells transduced with shIL-18 or shCTL (data are representative of two samples in three experiments performed in triplicate).

E–G  *In vitro* growth of M105 T-ALL cells in which IL-18Rα was silenced or not with shIL-18Rα/GFP or shCTL/GFP lentiviral vectors. Cells were transduced, sorted for GFP expression (E, day 0) and co-cultured with MS5 cells for 38 days. Quantification of GFP+ and GFP−/low cells at different time points (F). Black line, shCTL cells; grey line, shIL-18Rα cells. At day 38, T-ALL cells were sorted according to their GFP expression and IL-18Rα expression was quantified (G). Two independent experiments were performed.

Data information: Mann–Whitney non-parametric test was used for statistics.

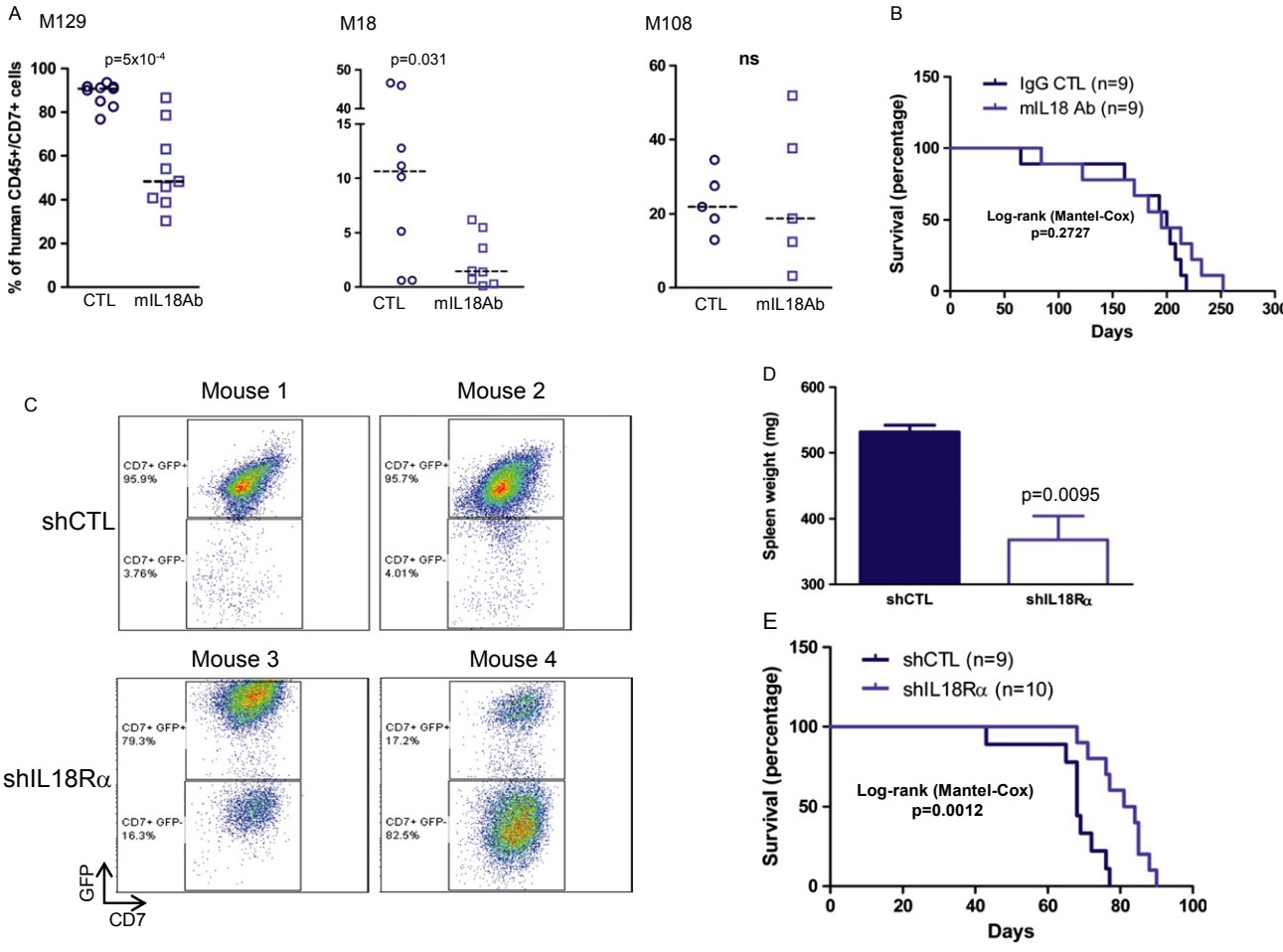

**Figure 4. Interfering with IL-18 activity delays leukaemia progression in xenografted mice.**

A   Proportion of human leukaemic blasts (CD45+CD7+ cells using anti-human CD45 and CD7 antibodies) in the bone marrow of NSG mice at week 6 after intravenous injection of $0.5$–$5 \times 10^3$ M129 (left panel), M18 (middle panel) or M108 (right panel) T-cell acute lymphoblastic leukaemia (T-ALL) cells that were pre-treated *in vitro* with mIL-18Ab or with isotype control Ab (CTL) for 2 weeks.

B   Survival curves of NSG mice transplanted with M129 T-ALL cells and shown in (A).

C   GFP expression in CD45+CD7+ leukaemic blasts recovered from NSG mice transplanted with M105 T-ALL cells that were transduced with shIL-18Rα/GFP (n = 10 mice) or shCTL/GFP vectors (n = 9 mice). Shown are results of four representative animals.

D   After euthanasia, spleens were recovered and weighed (mean ± s.d. of four mice transplanted with shCTL M105 T-ALL and six animals injected with shIL-18Rα M105 T-ALL).

E   Survival curves of mice transplanted with $5 \times 10^3$ shCTL or shIL-18Rα M105 T-ALL GFP-positive cells.

value of circulating IL-18 levels for T-ALL patients classified according to their EGIL phenotype.

# Discussion

We previously showed that T-ALL cells require stromal support for *in vitro* growth and that the NOTCH pathway is of particular importance for their maintenance (Armstrong *et al*, 2009), a finding that is supported by a recent study (Yost *et al*, 2013). Here, we used the T-ALL/MS5 stromal cell co-culture system to explore the role of the MEK/ERK pathway in T-ALL cell expansion. Unexpectedly, treatment with MEK inhibitors improved proliferation of about 70% of human T-ALL cell samples co-cultured with MS5 cells. This effect was not NOTCH dependent because MS5 cells do not express robust

levels of NOTCH ligand DL1 and NOTCH target genes were not activated upon incubation with MEK inhibitors. However, the finding that proliferation of T-ALL cells co-cultured with MS5-DL1 cells was only moderately modified following MEK inhibition suggests the existence of different T-ALL supporting niches. We then demonstrate that MEK inhibitors stimulate T-ALL cell proliferation through the induction of IL-18 secretion by MS5 cells.

These findings strongly support the hypothesis that cross-talk between stromal/microenvironmental cells and leukaemic blasts is crucial during T-ALL development (Nwabo Kamdje & Krampera, 2011). An emerging notion is that aberrant activation of signal transduction pathways in leukaemia cells can be induced by interactions with stromal cells of the bone marrow or thymus microenvironment (Konopleva & Andreeff, 2007; Lane *et al*, 2009). An altered microenvironment may even rupture the homeostasis and favour the early

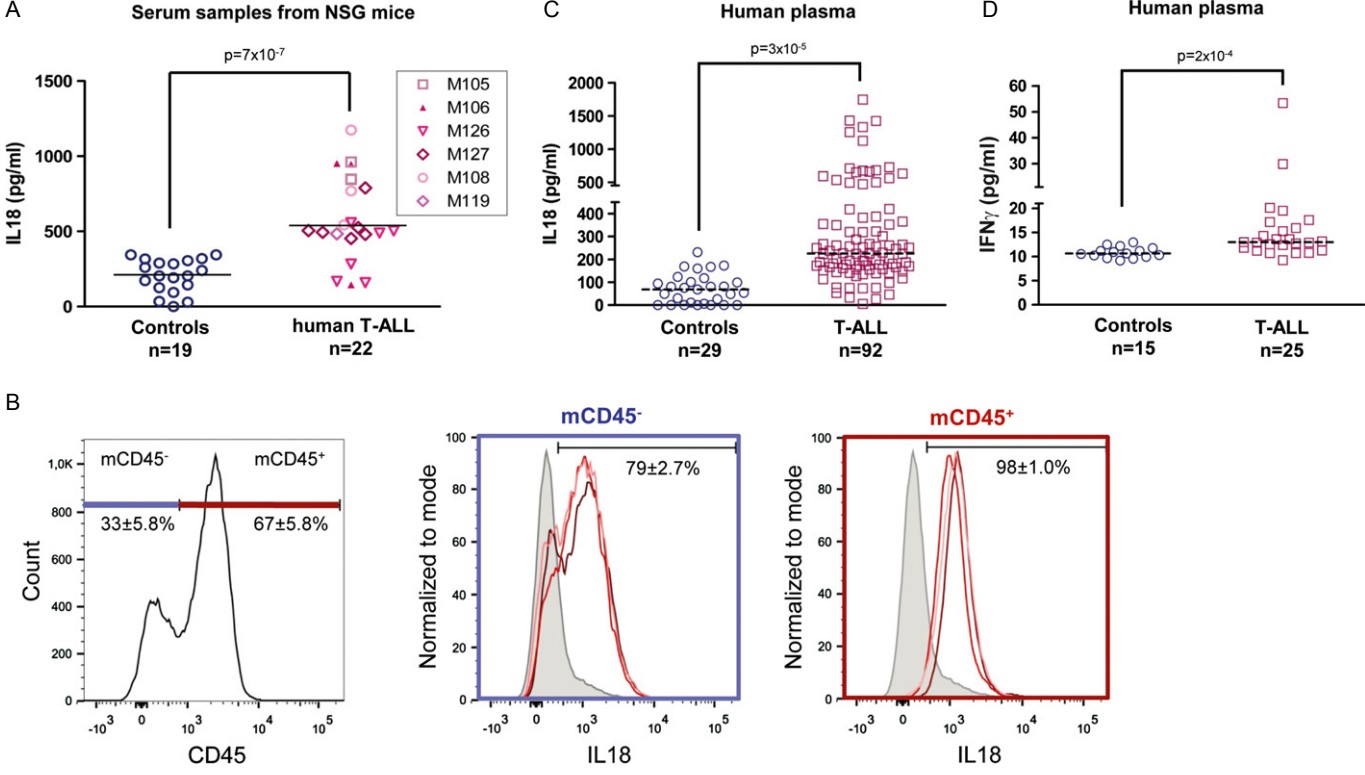

**Figure 5.   IL-18 production is increased in xenografted mice and in leukaemic patients.**

A      Circulating IL-18 expression levels in NSG mice injected with the indicated T-ALL samples (controls are non-transplanted 8- to 12-week-old NSG mice). Serum samples were collected at the time of euthanasia (6–12 weeks post-transplant) and IL-18 levels were measured by ELISA. Mann and Whitney non-parametric test was used for statistics.

B      Detection of IL-18 expression in BM from a tumour-free NSG mice. Medullary cells were recovered after flushing and crushing bones. Detection of IL-18 expression was measured by flow cytometry after permeabilization and anti-IL-18 labelling of BM cells.

C, D    Human IL-18 and IFN-γ levels in plasma samples from paediatric patients with T-ALL (IL-18: $n = 92$ and IFN-γ: $n = 25$) and healthy controls (IL-18: $n = 29$ and IFN-γ: $n = 15$). Statistical analyses were carried out using Student's *t*-test (C) or Mann–Whitney non-parametric test (D).

steps of transformation (Raaijmakers *et al*, 2010). Moreover, stromal cells can provide not only an adequate metabolic environment for leukaemic cells via activation of survival mechanisms, but also a protective niche to enhance resistance to chemotherapy (Zhang *et al*, 2012). Cytokines and growth factors secreted by stromal cells are suspected to exert a protective effect not only on the bulk of the leukaemic cell population, but also on the rare leukaemia initiating cells (LIC; Konopleva *et al*, 2009). In particular, it has been established that T-ALL cell proliferation and survival is regulated by several factors, including IL-7 and IGF-1 (Barata *et al*, 2001, 2004; Medyouf *et al*, 2011). These factors are likely to be produced by the microenvironment, particularly by bone marrow stromal cells (Barata *et al*, 2001, 2004; Scupoli *et al*, 2008). The increased expression of IL-18 in MEKi-treated MS5 stromal cells together with the detection of IL-18Rα/β in T-ALL primary cells and the finding that IL-18 incubation stimulates proliferation in more than half of the tested T-ALL samples strongly suggest that IL-18 also should be included among the microenvironmental signals supporting leukaemia development/progression.

IL-18 plays an important pro-inflammatory role (Mariathasan *et al*, 2004) and is involved in the induction of inflammatory mediators (Arend *et al*, 2008). Inflammation is implicated in the progression of different haematological or non-haematological

malignancies, and several pro-inflammatory factors, such as IL-6 and TNF-α, play a role in cancer development or progression (Liu *et al*, 2011; Reynaud *et al*, 2011). The B-ALL microenvironment is also described as inflammatory (Espinoza-Hernandez *et al*, 2001; Purizaca *et al*, 2012), further supporting the more general role of inflammation in acute lymphoblastic leukaemia. Moreover, elevated levels of IL-18 have been detected in solid tumours (Ye *et al*, 2007) similar to the levels we measured in T-ALL patients. Macrophages, chondrocytes and osteoblasts as well as medullary dendritic cells are known IL-18 producers (Arend *et al*, 2008), and here, we show that stromal/MSC cells also produce IL-18. In addition, IL-18 circulating levels were higher in NSG mice xenografted with human T-ALL cells in comparison with controls, arguing that IL-18 is induced in the environment following leukaemic blast spreading.

The role of IL-18 as a pro- or anti-tumour agent has been discussed in the literature and seems greatly dependent on the cancer cell type (Park *et al*, 2007). Our findings indicate that, in patients with T-ALL (and also in T-ALL xenograft models), IL-18 production is enhanced and it acts as a tumour cell proliferative factor. However, we cannot exclude the possibility that IL-18 might also drive the anti-tumour immune response *in vivo*, particularly through enhanced production of IFN-γ during the early stages of T-ALL development. Indeed, in some co-cultures, we observed that

**Table 2. Correlative analysis of IL-18 plasma levels and different clinical parameters in a cohort of 81 patients with T-cell acute lymphoblastic leukaemia (T-ALL) (n = 77)/T-lymphoblastic lymphoma (LBL; n = 4)**

| | T-ALL/T-LBL IL-18 plasmatic level (n = 81) | | Statistical significance ($c^2$ or Fisher's exact test) |
|---|---|---|---|
| | Upper level[a] (n = 40) | Low level[a] (n = 41) | |
| Age | | | |
| ≤ 10 years (%) | 22 (55) | 20 (49) | NS |
| > 10 years (%) | 18 (45) | 21 (51) | |
| M/F ratio | 3.0 | 2.7 | NS |
| Mediastinal involvement[b] + effusion[c] (%) | 3 (7.5) | 15 (37) | $P < 0.01$ |
| Tumour spreading[d] (%) | 29 (72) | 22 (54) | NS |
| CNS involvement (%) | 2 (5) | 3 (7) | NS |
| WBC (%) | | | |
| < 100 G/l | 21 (53) | 28 (68) | NS |
| ≥ 100 G/l | 19 (47) | 13 (32) | |
| European Group of Immunological Classification of Leukaemia (%) | | | |
| T I/II | 13 (32) | 11 (27) | NS |
| T III | 19 (48) | 25 (61) | |
| T IV | 6 (15) | 3 (7) | |
| Non-classified | 2 (5) | 2 (5) | |
| Cytogenetic abnormalities | (n = 35) | (n = 36) | NS |
| Yes (%) | 26 (74) | 23 (64) | |
| No (%) | 9 (26) | 13 (36) | |
| Response to prephase (%) | | | |
| Poor | 17 (42) | 11 (27) | NS |
| Good | 23 (58) | 30 (73) | |

[a]Based on the median secretion level; [b]chest X-ray assessment; [c]pleural and/or pericardial. [d]Spleen, liver, lymph nodes > 2 cm, kidney, testis, skin (at least two organs involved). Response to prephase was assessed after 7 days of steroid treatment (60 mg/m$^2$/day) + intrathecal methotrexate and defined as: poor ≥ 1 G/l peripheral blasts; good < 1 G/l peripheral blasts.

normal T cells, which are present in the samples isolated from newly diagnosed patients, proliferated in response to IL-18 and could overtake leukaemic blasts (data not shown), suggesting that some of the remaining immune cells also are sensitive to IL-18. Moreover, development of ICN1-driven mouse leukaemia is favoured in *IL-18−/−* mice in correlation with a reduction in the number of immunocompetent cells, indicating that the immune system is important for controlling T-ALL development (unpublished observations and; Schneider *et al*, 2010). IL-18 is indeed described as a regulator of the cytotoxic activity of NK and mature T cells and supports the differentiation and activation of different T-helper (Th) cell subsets, depending on the surrounding cytokine profile (Gracie *et al*, 2003). In combination with other interleukins, such as IL-12 and IL-15, IL-18 can be used to pre-activate NK cells with anti-tumour functions (Ni *et al*, 2012). Thus, targeting IL-18 in

patients with T-ALL needs to be carefully examined because it may also result in reduction of the immune response as previously suggested for other tumours (Park *et al*, 2007).

Our results show that mice xenografted with T-ALL cells in which IL-18Rα was silenced have reduced splenomegaly and better survival than animals injected with control T-ALL cells. This result supports the pro-leukaemia effect of stromal IL-18 in T-ALL, possibly through enhanced growth in the medullary microenvironment. However, as the white blood cell count was not particularly higher in high IL-18 patients compared to low IL-18 patients, the medullary niche could also offer a protective area for IL-18-responding patients and additional, unknown factors might be implicated in the stroma/leukaemic blast cross-talk. This is supported by the finding that IL-18 does not increase proliferation of primary T-ALL cells cultured without stromal cell support (not shown). In pathologies such as rheumatoid arthritis, IL-18 has been shown to promote migration, adhesion and activation of leucocytes through the release of different factors such as SDF1/CXCL12 and VCAM1/ICAM1 (Komai-Koma *et al*, 2003; Volin & Koch, 2011). The implication of IL-18 in these biological processes during T-ALL development/progression should be further explored.

There is some evidence for a role of the microenvironment in the outcome of acute leukaemia mainly due to the protective effect of the bone marrow niche on LICs against chemotherapeutic agents (Ayala *et al*, 2009). Although our data on patients' outcome according to the IL-18 secretion level need to be extended to a larger cohort and also to a supplementary and independent group of patients, they suggest that the microenvironment plays a role in T-ALL outcome via secreted factors. Dissecting further the clinical impact of microenvironmental factors in T-ALL will certainly be of a particular importance for future therapeutic strategies.

To summarize, here we show that T-ALL cell expansion is supported by the pro-inflammatory IL-18 cytokine secreted by bone marrow-derived stromal cells. These results highlight the implication of the microenvironment in human T-ALL progression and the relevance of studying the interactions between leukaemic blasts and feeder cells for identifying novel therapeutic strategies.

# Materials and Methods

### Patients and cell samples

Primary T-ALL cell samples were collected after obtaining informed consent by the patients or the patient's relatives in accordance with the Declaration of Helsinki and French ethics regulations. Blood cells were ficolled, immunophenotyped and directly used, or stored frozen. All patients were treated with an ALL-BFM-based strategy in the EORTC 58881 and 58951 trials (Clappier *et al*, 2010). The INSERM ethical committee (IRB000003888) approved the project (evaluation number 13-105-1). ICN1-induced mouse T-ALL cells were obtained as described (Gachet *et al*, 2013).

The choice of samples used for experiments was motivated by the availability of enough cells from individual patients and also to optimization of protocols, such as lentiviral transduction, on some, but not all, samples.

Plasma samples used for measuring IL-18 and IL-18BP levels were all from children (T-ALL cohort and controls). These patients are different from the ones used for functional experiments and

represent a retrospective cohort. Plasma samples from a subgroup of these T-ALL patients (those with the highest IL-18 level) and from a randomly chosen group of children controls were used for IFN-γ measurements.

## MS5 cells

Mouse stromal MS5 and MS5-DL1 (expressing the NOTCH ligand DL1) cells were described previously (Armstrong *et al*, 2009).

## Culture conditions

T-ALL cells ($100$–$250 \times 10^3$/well) were cultured as in (Armstrong *et al*, 2009). MEK pharmacological inhibitors (MEKi: U0126, PD98059, PD184352; Selleckchem/Euromedex, Souffelweyersheim, France) were diluted in DMSO and used at 20, 10 and 1 μM, respectively, or as otherwise indicated. MEKi, recombinant human IL-18 (100 ng/ml; Biovision/Clinisciences, Nanterre, France) and neutralizing anti-IL-18 antibodies (200 ng/ml; MBL/CliniSciences) were added in the culture medium 3 times/week. ICN1-induced mouse T-ALL cells were cultured as in Gachet *et al*, (2013). Growth was measured by gating on $CD45^+CD7^+$ blast cells using flow cytometry as in Armstrong *et al* (2009).

## Flow cytometry

Cells were stained with fluorescein (FITC)-, phycoerythrin (PE)-, PE-cyanin7 (PC7)- and allophycocyanin (APC)-conjugated mouse monoclonal antibodies (mAbs) specific for human CD45 (J33), CD7 (8H8.1) (1:50 dilution, Beckman Coulter; Roissy CDG, France and eBioscience, Paris, France), IL-18Rα and IL-18Rβ (FAB840P and FAB118P, 1:10 dilution; R&D Systems, Lille, France) and analysed using a FACSCalibur cytometer (BD Biosciences, Le Pont de Claix, France). For intracellular mIL-18 detection in murine BM, cells were permeabilized using BD Cytofix/Cytoperm™ (BD Biosciences) according to the manufacturer's protocol. Then, cells were stained with a purified rabbit anti-IL-18 (H-173) antibody (Santa Cruz Biotechnology, Dallas, TX, USA), and IL-18 expression was revealed by FACS using an anti-rabbit Alexa 647 antibody (Molecular Probes, Eugene, OR, USA). Cell sorting was performed using an Influx cell sorting cytometer (BD Influx system; BD Bioscience). ICN1-induced mouse T-ALL cells were monitored with an anti-mouse CD45 mAb (eBioscience).

## Lentiviral vectors and transduction of stromal cell lines and T-ALL primary cells

### IL-18Rα gene silencing in primary T-ALL cells

The *IL-18Rα* short hairpin RNA (shRNA, 5′-GCCTGTTCTTT CCGAGTCTTA-3′), the expression of which is driven by the H1 promoter, was subcloned in the pTRIP/ΔU3-MND-GFP vector as in previous studies (Kusy *et al*, 2010). The control hairpin sequence (sh*CTL*) targets the human hepatitis B virus. Human T-ALL cells were transduced with not concentrated G/IL-7SUx envelope pseudotyped vectors as in Gerby *et al*, (2010). Transduced cells were sorted based on the $CD45^+CD7^+GFP^+$ phenotype before use.

### ERK *and* IL-18 *gene silencing in MS5 cells*

The H1-shRNA cassettes against mouse *ERK1* and *ERK2* (kindly provided by Dr P. Lenormand, Institute of Signalling Developmental Biology and Cancer, Nice, France; Lefloch *et al*, 2008) were subcloned in the pTRIP/ΔU3-EF1a lentiviral vector. The same strategy was used to knock down mouse *IL-18* in MS5 cells with a shRNA against mouse *IL-18* (shRNA/mIL-18, 5′-CCTCTCTGTGAAGGATAGTAA-3′). Subconfluent MS5 cells were incubated with supernatants containing the vectors for 3 days, then carefully washed and a sample was harvested to quantify transduction by measuring GFP expression by FACS. Silencing efficiency was assessed by QPCR (IL-18) or by western blotting (ERK1/2). Control shCTL was as for T-ALL cells.

## Microarray analysis

RNA was extracted from MS5 cells incubated or not with 1 μM PD184352 for 7 days (three independent experiments = six samples in total) and was spotted on whole-genome arrays (mouse Gene1.0 ST; Affymetrix, Bucks, UK). First, we compared all treated and untreated samples to identify genes differentially expressed and then we crossed the data from the three independent experiments. Experimental details are provided in Supplementary Fig S4.

## Bioinformatics

### Quality assessments, normalization and statistics

Raw data were normalized using the Robust Multichip Algorithm (RMA) in Bioconductor R. Then, all quality controls and statistics were performed using Partek GS. Hierarchical clustering (Pearson's dissimilarity and average linkage) and principal component analysis were used. To find genes that were differentially expressed in treated and untreated samples, we applied a classical analysis of variance for each gene and carried out pairwise Tukey's *post hoc* tests between the groups. We then considered $P$-values $\leq 0.05\%$ and a $\geq 1.5$-fold change as significant for filtering and selecting differentially expressed genes.

### Functional analysis

Interactions, pathways and functional enrichment analyses were carried out using IPA (Ingenuity Systems, USA www.ingenuity.com). All microarray analysis data have been submitted to GEO Omnibus (accession number: GSE43623).

## Immunoblot analysis

Total proteins from T-ALL or MS5 cells were extracted with 1× lysis buffer, separated by 12% SDS–PAGE and transferred onto nitrocellulose membrane (GE Healthcare, Velizy-Villacoublay, France). Membranes were incubated with antibodies against phospho-p44/42 MAPK (ERK1/2) (Thr202/Tyr204) (4376; Ozyme, Saint Quentin Yvelines, France), p44/42 MAPK (ERK1/2) (4695; Ozyme) and β-actin (clone AC-15; Sigma-Aldrich). Proteins were detected with the ECL PLUS chemiluminescent substrate (GE Healthcare).

## Electrophoretic mobility shift assays (EMSA)

Nuclear extracts were prepared, and 15 μg of proteins was analysed for DNA binding activity using the HIV-LTR tandem κB

**The paper explained**

**Problem**

T-cell acute leukaemia (T-ALL) is an aggressive haematological disorder that accounts for 15–25% of acute lymphoblastic leukaemia. Outcome varies depending on the age of onset, and generally, adults are much less well cured than young patients. Moreover, relapses have a very bad prognosis, independently of the patient's age. There is thus a great need for complementary therapeutic approaches to increase patients' remission. T-ALL development is characterized by the accumulation of genetic accidents that are considered as targets for dedicated therapies. Moreover, there is growing evidence that interactions between leukaemic cells and their microenvironment promote leukaemia development and progression. This last observation opens new areas for novel therapeutic developments.

**Results**

Using the only culture system available for growing *ex vivo* T-ALL cells isolated from patients, we uncover the role of the pro-inflammatory cytokine interleukin 18 (IL-18) in supporting human T-ALL propagation. We demonstrate that stromal cells that are part of the medullary microenvironment and are included in the culture system can secrete IL-18, upon treatment with inhibitors of the MEK/MAP kinase pathway. Loss of function of IL-18 in stromal cells or knocking down of IL-18 receptors (IL-18R) in T-ALL blasts seriously interferes with leukaemia cell growth in culture. Using mice xenografted with human T-ALL cells, we show that leukaemia development induces IL-18 production in these animals. Moreover, in mice xenografted with T-ALL cells in which IL-18R expression is reduced, leukaemia development is delayed and animals' survival extended. Finally, we show that IL-18 plasmatic levels are also increased in paediatric patients with T-ALL.

**Impact**

This study outlines the critical interaction between leukaemia cells and the medullary microenvironment and the positive effect on tumour growth of microenvironment secreted factors. It highlights a new mechanism of leukaemia progression and thus may represent an additional entry for therapies.

oligonucleotide as a probe for NF-kB (Jacque *et al*, 2005). Samples were then resolved by electrophoresis using acrylamide/bisacrylamide/TBE gels.

## IL-18 and IL-18BP plasma levels

IL-18 and IL-18BP plasma levels were measured using ELISA kits (mouse IL-18 Platinum ELISA, BMS618/2, human IL-18 Platinum ELISA, BMS267/2CE; eBioscience and human IL-18BP ELISA Kit, ab100559; Abcam, Paris, France) following the manufacturer's instructions. Free IL-18 levels were calculated using IL-18 and IL-18BP levels as shown in Novick *et al*, (2001).

## Real-time quantitative RT–PCR

RNA was isolated with TRIZOL (Life Technologies, Saint Aubin, France), reverse-transcribed using random hexamers and the Superscript RT kit (Invitrogen) according to the manufacturer's instructions. The products were used as template for real-time PCR using the Power SYBR Green PCR Master mix or the TaqMan Universal PCR MasterMix No AmpErase UNG and the 7300 fast real-time PCR system (Life Technologies). Raw data were obtained in terms of Ct values and normalized to the Ct values of the housekeeping genes *GAPDH* or *β2m*. The primer sequences are in Supplementary Table S1.

## Animals

NOD.CB17-Prkdc(scid) (abbreviated NS) and NOD.Cg-Prkdc (scid)Il2rg(tm1Wjll)/SzJ (abbreviated NSG) mice (The Jackson Laboratory, Bar Harbor, ME, USA) were housed in pathogen-free animal facilities at CEA, Fontenay-aux-Roses, France. Mice were irradiated with 2.5–3 Gy (IBL 637 CisBio International, France; dose rate: 0.61 Gy/min) and anesthetized with isoflurane before intravenous injection of human leukaemic cells. Experimental procedures were performed in compliance with the French Agriculture Ministry and local ethics committee regulations (Authorization number 12-015). Mice were euthanized when they reached endpoints set to meet accepted animal care guidelines. Leukaemia progression was monitored using the anti-human PE-CD45 and PC7-CD7 mAbs (Beckman Coulter and eBioscience).

## Statistical analyses

Statistical significance of comparisons was determined using the Mann and Whitney non-parametric test. Disease-free survival of mice was estimated using the Kaplan–Meier method and subgroups were compared using the log-rank test. Survival of patients was measured using a Cox-regression model.

**Supplementary information** for this article is available online: http://embomolmed.embopress.org

## Acknowledgements

The authors would like to thank the donors for allowing the use of their cells for research, and the clinical teams at the Hôpital A Trousseau and R Debré in Paris and IHOP in Lyon who collected the samples, particularly Dr Y Mekki at the Hospices Civils de Lyon for collecting the plasma samples. We acknowledge F De Toni and L Casteilla (Stromalab, Toulouse) for human MSC cells and M Gaudry (U1016, Paris) for shERK1/2 supernatants. We are grateful to I Naguibneva, L Renou, A Ben Youcef, E Soler and PH Roméo for their critical comments. Cell sorting and analysis was done at the IRCM flow cytometry facility supervised by J Baijer. Mouse work was greatly facilitated by C Joubert and J Tilliet from the IRCM animal facility. Microarray analysis was performed at the GENOM'IC platform, Institut Cochin, Paris. E Andermacher edited the manuscript. This work was supported by INSERM, CEA, Université Paris Diderot and Université Paris Sud, Institut National du Cancer (INCA, Network 2008), Cancéropôle Ile de France, StemPole, the Ligue Nationale contre le Cancer (LNCC, équipe labellisée) and the Association Laurette Fugain. SP and XC are supported by INCA. BG had fellowships from LNCC and Société Française d'Hématologie.

## Author contributions

BU, SP, BG, CLW, JG, FA, JC, DP, CBG, JG and CD performed experiments. BU, SP, XC, FD, VB, FB and FPf analysed data. TL, AB, JLP, PB, JG and FB provided important reagents. BU, FB, FPo and FPf conceived the project and wrote the manuscript. All authors critically reviewed the manuscript.

## Conflict of interest

The authors declare that they have no conflict of interest.

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
