## [Review Process File · EMBO Molecular Medicine]

Interleukin-18 produced by bone marrow derived stromal cells supports T cell acute leukemia progression

Benjamin Uzan, Sandrine Poglio, Bastien Gerby, Ching-Lien Wu, Julia Gross, Florence Armstrong, Julien Calvo, Xavier Cahu, Caroline Deswarte, Florent Dumont, Diana Passaro, Corinne Besnard-Guérin, Thierry Leblanc, André Baruchel, Judith Landman-Parker, Paola Ballerini, Véronique Baud, Jacques Ghysdael, Frédéric Baleyrier, Françoise Porteu and Françoise Pflumio

Corresponding author: Françoise Pflumio, UMR967 INSERM/CEA

Review timeline:

Submission date:	26 July 2013
Editorial Decision:	31 July 2013
Appeal:	22 August 2013
Editorial Decision:	23 September 2013
Revision received:	21 January 2014
Editorial Decision:	13 February 2014
Revision received:	25 February 2014
Accepted:	03 March 2014

Transaction Report:

Editor: Roberto Buccione

1st Editorial Decision

31 July 2013

Thank you for the submission of your manuscript "Interleukin-18 produced by bone marrow derived stromal cells supports T cell acute leukemia propagation".

I have now had the opportunity to carefully read your paper and the related literature and I have also discussed it with my colleagues. I am afraid that we concluded that the manuscript is not well suited for publication in EMBO Molecular Medicine and have therefore decided not to proceed with peer review.

You find that inhibition of MEK promoted growth of xenografted or human primary leukemic cells in co-culture with stromal cells and that inhibition of MEK specifically in stromal cells promoted T-ALL cell proliferation. You also find IL-18 as major secreted factor and confirm that T-ALL cells have IL-18 receptors and a portion of samples are stimulated by IL-18, possibly via NFκB. We appreciate that an IL-18 blocking body decreased T-ALL growth in xenografted mice and that IL-18 levels were increased both in xenografted mice and in patients. Finally, you observed a better outcome for low IL-18 in a specific clinical subset.

Although we acknowledge the potential interest of your manuscript, we feel that it is better suited for a specialty venue. We are in fact, not persuaded that at this stage your manuscript provides the striking conceptual advance and mechanistic insight we would like to see in an EMBO Molecular Medicine article.

I am sorry that I could not bring better news.

Appeal

22 August 2013

I read carefully your message and I would like to make an appeal on your decision.

You are telling that you are not "persuaded that our manuscript provides the striking conceptual advance and mechanistic insight" you would like to see in EMM. I am surprised by this comment. T-ALL are usually studied through the acquisition of genetic defects. The whole scientific community working on T-ALL is sure all the abnormalities lie inside of leukemic cells. We and a few other labs have found that in fact T-ALL, as other types of leukemia, are dependent on microenvironmental factors and here in this work we point to a new and original factor, the pro-inflammatory cytokine IL18, and we conclusively show that it is implicated into T-ALL progression. We provide results with human T-ALL samples and with mouse transgenic cells. We provide data on patients with results pointing at IL18 as a new prognostic marker for a subtype of T-ALL patients.

I am thus asking you if you could reconsider your position concerning our work and send it out for review.

I thank you in advance for your help.

2nd Editorial Decision

23 September 2013

Thank you for the submission of your manuscript to EMBO Molecular Medicine. We have now heard back from the three Reviewers whom we asked to evaluate your manuscript.

You will see that while the Reviewers are supportive of your work, they express a number of complementary concerns that prevent us from considering publication at this time. I will not dwell into much detail, as the evaluations are detailed and self-explanatory and will just mention a few main points.

Reviewer 1 raises an important technical issue that potentially and ultimately impinges on the interpretation of the results. S/he suggests that it is of pivotal importance to know the levels of IL18BP to draw conclusions on the importance of IL18.

Reviewer 2, while recognising the use of primary samples, would like to be convinced that there was no selection bias for these samples. S/he also notes that expression profiling was done only after prolonged MEKi incubation and as a consequence would like to better understand the kinetics of IL18 induction and the relationship with the effects of MEKi *in vivo*. Reviewer 2 lists other important issues that require your action.

Reviewer 3 is concerned that there is no evidence that there are IL18-producing cells in the bone marrow and suggests that this aspect be explored. S/he would also like you to better analyse the reasons behind the fact that a large fraction of T-ALL cells do not benefit from MEKi treatments, and suggests some strategies to that effect. Reviewer 3 also feels that a better control should be used in analysing IL18 plasma levels in childhood T-ALL cases. Finally, this Reviewer also feels that the clinical relevance of the data on the prognostic value of IL18 levels is far from convincing. I agree with this assessment. Ideally, to provide data from a larger cohort of patients would strengthen this aspect and increase the impact of the manuscript. If this is not possible, and as the Reviewer mentions, the conclusions would remain speculative and would need to be toned-down (including in the Abstract). Reviewer 3 also lists other important issues that require your action.

Considering all the above, while publication of the paper cannot be considered at this stage, we would be pleased to consider a substantially revised submission, with the understanding that the Reviewers' concerns must be fully addressed, with additional experimental data where appropriate and that acceptance of the manuscript will entail a second round of review.

Please note that it is EMBO Molecular Medicine policy to allow a single round of revision only and that, therefore, acceptance or rejection of the manuscript will depend on the completeness of your responses included in the next, final version of the manuscript.

As you know, EMBO Molecular Medicine has a "scooping protection" policy, whereby similar findings that are published by others during review or revision are not a criterion for rejection. However, I do ask you to get in touch with us after three months if you have not completed your revision, to update us on the status. Please also contact us as soon as possible if similar work is published elsewhere.

I look forward to seeing a revised form of your manuscript as soon as possible.

Referee #1 (Remarks):

Major point. The data that there is a role for IL 18 in T cell ALL is convincing but it is only half of the story. A role for IL 18 can only be considered unless you know the production and effect and levels of IL 18 binding protein. The affinity of the IL 18BP is far greater than that of the soluble IL 18 receptor. Many papers that deal with a pathological role for IL 18 measure both IL 18 and IL 18BP and then calculate the level of free IL 18. The present study needs to go back to all the data measuring IL 18, whether in vivo in patients or in vitro from cells and determine the level of IL 18 binding protein. Please read the papers dealing with free IL 18. Your paper will be all the better if you know both IL 18 and IL 18BP levels. For example, how do you know that the reason some patients do better than others is not due a high level of IL 18IL 18BP?

Minor points. The Elisa for IL 18 is strange in that the level in healthy persons is rather high. The MBL Elisa has the standards for healthy persons. Also I cannot find the concentration of IL 18 added to the cultures.

Referee #2 (Remarks):

Uzan et al. use an in vitro culture system for primary human T-ALL cells to explore the unexpected observation that addition of MEKi to the culture system stimulates the growth of a significant subset of T-ALLs. By expression profiling and other means, they link this effect to upregulation of IL-18 expression in MS5 cells, and then perform additional in vitro studies with in vivo correlates to implicate IL-18 as a factor that enhances T-ALL growth/survival. Finally, they provide correlative data suggesting that circulating IL-18 levels are associated with a poor prognosis in human T-ALL.

These are novel findings with interesting translational and clinical implications. Certain aspects of the data and its interpretation are imprecise or unclear, as follows.

1. It is to the authors' credit that they have done the work described here with primary samples, but the selection of samples for the studies seems to be quite ad hoc. Some are used for some studies, others for others; some sense of why particular tumors were used for particular studies is needed to allay fears of data selection.

2. Table 1. This table presents information on 22 primary lines, 14 of which appear to grow better in the presence of MEKi, either with MS5 or MS5-DLL1 feeders. However, no explanation is provided about how growth was quantified; what does +, ++, +++, etc., actually mean and how was it measured?

3. Table 2. It's interesting that mediastinal disease correlates with low IL-18 levels, since patients with mediastinal disease are more likely to have lymphomatous presentations with less marrow involvement. Is marrow involvement related to IL-18 levels? More mechanistically, do T-ALL cells stimulate IL-18 production by MS5 cells, independent of MEKi treatment? This is suggested by changes in IL-18 levels in T-ALL engrafted mice. And is there any past work describing IL-18 production by marrow-resident cells?

4. The authors show that MEKi results in upregulation of IL-18, but the profiling is done after prolonged drug incubation (7 days). What is the kinetics of IL-18 induction? Are these kinetics consistent (or inconsistent) with possible effects of MEKi *in vivo*?
5. The EGIL cortical phenotype appear to be over-represented in the low IL-18 group; does this imply that the Notch-on phenotype is also associated with low IL-18?
6. The statement in the discussion that the "IL-18 circulating level might neutralize the previously reported prognostic impact of T-ALL immunophenotype" requires clarification.

Referee #3 (Comments on Novelty/Model System):

The manuscript by Uzan et al demonstrates that, contrary to what one would expect from the fact that some T-ALL cases present MEK-Erk pathway activation, the use of MEK inhibitors enhances the growth of human T-ALL primary samples co-cultured with stromal cells. This is because apparently MEK inhibitors induce the production of IL18 in the stromal cells.

The authors combine clearly well and rigorously performed *in vitro* and *in vivo* xenotransplant experiments to show that IL18 and its receptor (expressed in T-ALL cells) promote leukemia proliferation/progression. In addition, they find that IL18 plasma levels are increased in T-ALL xenotransplanted mice and in patient samples as compared to healthy controls. Finally, they try to explore the potential prognostic value of their observations.

Overall, the experiments are adequately performed and the authors make use of a diverse array of very appropriate methodologies, including some very challenging ones, such as transducing primary T-ALL cells, which is technically extremely difficult.

The novelty of the work is not restricted to the surprising positive effect of MEK inhibitors towards T-ALL cells - an observation of major importance given that these could be potential tools for T-ALL treatment in light of the knowledge that some T-ALL cases present constitutive activation of this pathway. The identification of IL18 as a new player in T-ALL pathophysiology is exciting and novel.

Whether IL18 levels have prognostic value should be, in my opinion, better evaluated. Nevertheless, the clinical impact of these studies may be considerable: not only because of the unexpected impact of MEK inhibitors in T-ALL cells but also because IL18 is certainly a potential clinical target by using neutralizing antibodies against the cytokine.

Referee #3 (Remarks):

The manuscript by Uzan et al demonstrates that, surprisingly, the use of MEK inhibitors enhances the growth of human T-ALL primary samples co-cultured with stromal cells. This is because apparently MEK inhibitors induce the production of IL18 in the stromal cells. Combining *in vitro* and *in vivo* xenotransplant experiments the authors then show that IL18 and its receptor (expressed in T-ALL cells) promote leukemia proliferation/progression. In addition, they find that IL18 plasma levels are increased in T-ALL xenotransplanted mice and in patient samples as compared to healthy controls. Finally, Uzan et al. identify a subgroup of T-ALLs, with low IL18 levels and EGIL TIII stage, that appear to have good prognosis as compared to the remaining.

This is an exciting, highly novel, thought-provoking study, which is of potential clinical relevance. Altogether I find the manuscript of considerable interest, identifying IL18 as a new player in T-ALL and providing substantial evidence supporting the importance of the leukemic milieu for T-ALL disease progression.

Despite these clear qualities, there are several aspects that in my view raise concern or that could be improved and clarified.

1. Although the title of the manuscript alludes to the fact that IL-18 produced by bone marrow-derived stromal cells supports T-ALL progression, the authors don't actually show any *in vivo* evidence, from mice and if possible from humans, that there are IL18 producing cells in the BM. They should either mention previous studies in which this has been demonstrated (if any) or, in their absence, perform these critical analyses in BM biopsies.

2. Given the significant differences in current treatment efficacy (and the possibility that these may reflect biological peculiarities) between children and adult T-ALL patients, the authors should indicate in Table S2 each patient's age and immunophenotype. Moreover, they should clearly identify the 2 age groups in the Methods and main text. For example, in the Methods they refer that all patients were treated in the EORTC58881 and 58951 trials. This suggests that all cases in the present study are pediatric. However, the controls are sometimes adult healthy individuals, which hints that some of the T-ALL cases may be adults. This should be clarified in the manuscript.

3. The effect of the MEKi could not be properly evaluated in those samples that did not grow in vitro. Therefore, it seems more logical that the percentage of samples in which MEK has a stimulatory effect should be calculated in relation to those that grew in vitro (n=22) and not the overall number of samples. On the other hand, some of the samples presented in Table 1 are from the same patient (namely M18/M18x and M105/M105x). They are not really independent, even if they represent different "states" (ex vivo versus xenotransplanted). Therefore, it seems more appropriate that, for the calculation of the percentage of MEKi-responsive samples the total number of samples analyzed is defined as 20 (not 22). The same rationale should be applied (and it seems to have been in some instances, e.g. in the legend to fig. 1) in other parts of the manuscript.

4. Why is a large fraction of T-ALL cells not benefiting from MEKi treatment?

Given the generally anti-apoptotic and pro-oncogenic role of MEK-Erk pathway, and the fact that a significant fraction of T-ALL cases display constitutive MEK-Erk pathway activation, it is possible that the MEKis have a negative effect on some of the T-ALL cells themselves, especially those cases with the highest levels of MEK-Erk pathway activation. This could explain why not all T-ALL samples co-cultured in stroma benefited from MEKi treatment, since the positive effect that the MEKis had via stromal cells may have been counterbalanced by a direct negative effect on T-ALL cells. Thus, the authors could: a) evaluate whether sensitivity to the MEKis correlated with MEK-Erk activation in T-ALL cells; and b) test the effect of MEKis on T-ALL cells directly (no co-culture).

An alternative possibility is that there are T-ALL cases with low/absent IL18R surface expression, in which case these samples should be insensitive to the MEKis. IL18R levels should be correlated with sensitivity to MEKi in the co-cultures. Likewise, responsiveness to rhIL18 should be correlated with IL18R surface expression in T-ALL cells.

Addressing these points would significantly expand the characterization and understanding of the unexpected (and very interesting) impact of MEKis on T-ALL cells, which may be of considerable therapeutic relevance.

6. In Figure 3B the authors show that exogenous IL18 leads to NF-kappaB activation in T-ALL cells. Does the same happen in MEKi-treated T-ALL cells in co-culture with stroma? Most importantly, is NF-kappaB activation relevant for IL18-mediated effects, i.e. if the authors pre-treat T-ALL cells with an NF-kappa B inhibitor and then co-culture them with stromal cells alone or in the presence of IL18, is there a blockade in the effect of the cytokine? Addressing this question would strengthen significantly the relevance of the link between IL18 and NF-kappa B.

7. In Figure 5B, childhood T-ALL cases are compared with a mix of healthy pediatric and adult controls regarding IL18 plasma levels. To avoid the possible criticisms that there may be age-related differences in IL18 expression in healthy controls that could create a bias in the analysis, the authors should remove the adult samples and compare strictly the same age groups.

8. The actual clinical relevance of the data on the prognostic value of IL18 levels is questionable. There is a trend for better DFS in IL18 low patients (fig. 5D), but this trend is far from significant. Unless the authors provide further evidence from another, larger cohort of patients that allows for the actual identification of statistically significant differences, the data will remain speculative. Likewise, the authors should avoid using the analysis presented in panel 5E to speculate that "Also, the low IL18 group seemed to have a better outcome than the high IL18 group, whatever the EGIL phenotype (Figure 5E-F)." (page 9). The data presented in panel 5E do not add to fig.5D in terms of IL18 prognostic value per se and therefore do not allow to formally extract any conclusions on IL18 prognostic value.

Also, I have strong doubts as to whether the claim that "Combined with the EGIL stage, high IL18 plasma levels identified a patients' subset with poor outcome" (abstract). This statement does not appear to be corroborated by the data in fig. 5. If anything, the data in panel 5E identify a patient

subset with low plasma levels and EGIL TIII stage that present good prognosis. However, this is also arguable, because the authors compare EGIL TIII versus not TIII within the IL18 low-expressing cases. In such way, the prognostic variable is the EGIL stage (measured within a defined T-ALL subgroup with low IL18 levels), not IL18. If the authors are to claim that the combination of the two discrete variables (IL18 levels and EGIL stage) is of prognostic value, it seems to me that they must compare the four groups in a single statistical analysis (i.e. Hi IL18-EGIL TIII x Hi IL18-EGIL non TIII x Lo IL18-EGIL TIII x Lo IL18-EGIL non TIII). Otherwise, the clinical importance of their analyses will be limited.

Minor

1. There are typos in the EMSA section of the Methods.
2. Figure 5 legend, panel D: "(...) and low (IL18 levels above the median(...)). "above" should be corrected to "below" or "under". Also, in page 9, last sentence, the reference to figure 5E-F is incorrect. There is no panel F in this figure.
3. Figure 3C is somewhat "overcrowded" and difficult to follow. Similar to other panels, an inset with the data on day 28 would help making the results easier to read.
4. Statistics for Figure S6 are missing.

1st Revision - authors' response

21 January 2014

Referee #1

Major point.

The data that there is a role for IL 18 in T cell ALL is convincing but it is only half of the story. A role for IL 18 can only be considered unless you know the production and effect and levels of IL 18 binding protein. The affinity of the IL 18BP is far greater than that of the soluble IL 18 receptor. Many papers that deal with a pathological role for IL 18 measure both IL 18 and IL 18BP and then calculate the level of free IL 18. The present study needs to go back to all the data measuring IL 18, whether in vivo in patients or in vitro from cells and determine the level of IL 18 binding protein. Please read the papers dealing with free IL 18. Your paper will be all the better if you know both IL 18 and IL 18BP levels.

According to the reviewer's interesting comment we have measured the levels of IL18BP in plasma from 57 patients and 20 controls, all children. These results show that IL18BP is also increased in T-ALL as total IL18. We added a novel figure showing the relation of IL18BP and total IL18 concentrations in T-ALL and in controls (see **Figure S8A**). Also we have calculated the levels of free IL18 based on IL18BP and total IL18 levels according to *Novick, Cytokine, 2001*. We show that free and total IL18 are perfectly correlated (see **Figure S8B**). According to these results we have kept the levels of total IL18 in Figure 5 (but we implemented with data from additional patients) and we included in the manuscript a sentence outlining the fact that IL18BP is increased and that total and free IL18 are correlated (page 8-9).

Minor points.

The Elisa for IL18 is strange in that the level in healthy persons is rather high. In the cohort of the originally submitted manuscript, we had pooled results from healthy children and adults. These results have now been modified to show only healthy children (n=29) that are in fact better controls for our paediatric T-ALL cohort (as suggested by reviewer 3, in his comment #7). IL18 levels in such controls vary from <9pg/mL (detection limit of the ELISA assay) to 231pg/mL with median level of 68pg/mL and mean±SD=71±12pg/mL. We have found in the literature that normal levels of IL18 in adults are 64±17pg/mL (*Novick, Cytokine, 2001*) or in more recent works ranging from 80 to 120pg/mL (*Dinareello, Frontiers in Immunology, 2013*). Thus our mean levels are similar to previously published levels. Their range may be more heterogeneous than the published values

probably because they are measured in children that may still have a immature immune system (low IL18) but also a high propensity to viral/ bacterial infections (high IL18).

Also I cannot find the concentration of IL 18 added to the cultures. The concentration of IL18 (100ng/mL) in the cultures is included in the material and methods section, page 13.

Referee #2 (Remarks):

These are novel findings with interesting translational and clinical implications. Certain aspects of the data and its interpretation are imprecise or unclear, as follows.

We thank the reviewer for his encouraging comments and we'll try to answer the best we can to his questions.

1. *It is to the authors' credit that they have done the work described here with primary samples, but the selection of samples for the studies seems to be quite ad hoc. Some are used for some studies, others for others; some sense of why particular tumours were used for particular studies is needed to allay fears of data selection.*

The work we present has been started a few years ago. Therefore some of the samples used in the past were not available during the entire course of our study due to limited numbers of cells from every patient. We've done certain experiments with numerous samples (such as the co-cultures +/- MEKi) as we were testing samples as they arrived in the laboratory and we only performed cultures with IL18, or transduction experiments on more focused/limited number of leukaemia, due to cell number limitations and optimization only on limited samples. We now precise this point into the Material&methods section (page 13).

2. *Table 1. This table presents information on 22 primary lines, 14 of which appear to grow better in the presence of MEKi, either with MS5 or MS5-DLL1 feeders. However, no explanation is provided about how growth was quantified; what does +, ++, +++, etc., actually mean and how was it measured?*

We apologize to the reviewer because he may not have seen the legend of Table 1 (page 25) where it is indicated "CD45+CD7+ leukemic cells were counted at day 28 and (+), (++) , (+++), (++++), (+++++) indicate, respectively, a less than 10-fold, between 10- and 50-fold, 50- and 250-fold, 250- and 1000-fold and higher than 1000-fold number increase compared to the original number of seeded cells. 0, no cell was recovered after culture." Leukemic cells were counted using gating on CD45+CD7+ blastic cells by Flow cytometry as indicated now in the Material&methods section (page 13).

3. *Table 2. It's interesting that mediastinal disease correlates with low IL-18 levels, since patients with mediastinal disease are more likely to have lymphomatous presentations with less marrow involvement. Is marrow involvement related to IL-18 levels?*

All the patients with leukemia have a marrow infiltration with very high levels of blast cells detected irrespective of their mediastinal disease. Thus we could not find a relation between IL18 levels and marrow involvement.

More mechanistically, do T-ALL cells stimulate IL-18 production by MS5 cells, independent of MEKi treatment? This is suggested by changes in IL-18 levels in T-ALL engrafted mice. And is there any past work describing IL-18 production by marrow-resident cells?

We measured IL18 production by MS5 cells in contact with T-ALL cells and could not find any increased production compared to cultures without T-ALL. This result indicates either that such feeder cells do not reliably reproduce the physiological components of the BM niche and/or that the IL18-stimulation by T-ALL relies on another cell type. It is known that IL18 is mainly produced by dendritic cells and monocytes, and that mesenchymal cells and endothelial cells retain the pro-IL18 precursor that can be processed after cells die by neutrophil proteases (review in *Dinarelli, Frontiers in Immunology, 2013*). It is possible that *in vivo* T-ALL participates into IL18 production by directly stimulating monocytes and indirectly killing BM stromal/endothelial cell components during the medullary infiltration process. We have checked whether *in vivo* we could detect IL18 in mouse BM-resident cells. For this we have perfused T-ALL infiltrated-NSG mice with PBS to

circumvent blood contamination. Bones were recovered and crashed, extracted cells were permeabilized, labelled with anti-mouse IL18 antibodies and analysed by flow cytometry. The results show that mouse CD45⁺ and CD45⁻ cells do produce IL18. The results have been included in Figure 5 (as Fig5B) and are commented in the results section (page 8).

These results confirm past work describing IL18 production by marrow-resident cells, such as dendritic cells isolated from normal BM, that are capable in vitro to produce IL18 (Stoll, *Eur J Immunol*, 1998; Sakaki, *BBRC*, 2013) and by osteoblastic cells (Udagawa, *JEM*, 1997). IL18 expression has also been observed in BM of patients with pathologies such as arthritis (Maeno, *Arthritis Rheum*, 2004).

4. *The authors show that MEKi results in upregulation of IL-18, but the profiling is done after prolonged drug incubation (7 days). What is the kinetics of IL-18 induction? Are these kinetics consistent (or inconsistent) with possible effects of MEKi in vivo?*

We have done a kinetic measurement of IL18 production by MS5 cells, following MEKi treatment in vitro. The results are shown in **Figure 1-reviewer 2** (Figure removed as requested by authors) and indicate that IL18 production increases progressively until 7 days of treatment.

We do not understand the comments regarding MEKi in vivo since we never injected MEKi into mice. In the experiments we have performed with T-ALL in vivo (Figure 1C), cells had been pre-treated in vitro with MEKi during 1 to 2 weeks and injected afterwards into mice.

5. *The EGIL cortical phenotype appear to be over-represented in the low IL-18 group; does this imply that the Notch-on phenotype is also associated with low IL-18?*

Unfortunately we do not have the systematic data on NOTCH mutation status of the patient samples we used for the IL18 patient plasma measurement (Table 2) as this is a retrospective analysis on patients that were followed up several years before our study was started. However as EGIL cortical T-ALL are supposed to correspond to normal cortical T cell differentiation stages, it is to be expected that NOTCH will be on in these samples. This point shall definitively be addressed in future IL18 measurements in plasma of newly diagnosed patients.

6. *The statement in the discussion that the "IL-18 circulating level might neutralize the previously reported prognostic impact of T-ALL immunophenotype" requires clarification.*

We have changed this statement in the discussion, as it was not clear enough. What we meant is that "IL-18 circulating level may influence prognostic of T-ALL patients classified according to their EGIL phenotype."

Referee #3 (Remarks)

1. *Although the title of the manuscript alludes to the fact that IL-18 produced by bone marrow-derived stromal cells supports T-ALL progression, the authors don't actually show any in vivo evidence, from mice and if possible from humans, that there are IL18 producing cells in the BM. They should either mention previous studies in which this has been demonstrated (if any) or, in their absence, perform these critical analyses in BM biopsies.*

This comment has been addressed in the responses to reviewer 2. In brief, we now show results in **Figure 5B** indicating expression of IL18 in BM-resident hematopoietic and non-hematopoietic cells.

2. *Given the significant differences in current treatment efficacy (and the possibility that these may reflect biological peculiarities) between children and adult T-ALL patients, the authors should indicate in Table S2 each patient's age and immunophenotype. Moreover, they should clearly identify the 2 age groups in the Methods and main text. For example, in the Methods they refer that all patients were treated in the EORTC58881 and 58951 trials. This suggests that all cases in the present study are pediatric. However, the controls are sometimes adult healthy individuals, which hints that some of the T-ALL cases may be adults. This should be clarified in the manuscript.*

We apologize that the writing was not clear enough for the reviewer to understand that the T-ALL patients we tested in Figure 5 are paediatric patients only. However it is true that the controls were a mix of adults and children plasma. In fact we had found similar IL18 levels in healthy adults and children explaining why we had pooled these data. We have now retrieved the adult healthy control values from Figure 5C and kept only the healthy children values (n=29), as suggested by the

reviewer. This is now specified in the manuscript in Material&methods section.

3. *The effect of the MEKi could not be properly evaluated in those samples that did not grow in vitro. Therefore, it seems more logical that the percentage of samples in which MEK has a stimulatory effect should be calculated in relation to those that grew in vitro (n=22) and not the overall number of samples. On the other hand, some of the samples presented in Table 1 are from the same patient (namely M18/M18x and M105/M105x). They are not really independent, even if they represent different "states" (ex vivo versus xenotransplanted). Therefore, it seems more appropriate that, for the calculation of the percentage of MEKi-responsive samples the total number of samples analyzed is defined as 20 (not 22). The same rationale should be applied (and it seems to have been in some instances, e.g. in the legend to fig. 1) in other parts of the manuscript.*
We have changed the text in the manuscript in accordance of the reviewer comments (page 5).

4. *Why is a large fraction of T-ALL cells not benefiting from MEKi treatment?*

Given the generally anti-apoptotic and pro-oncogenic role of MEK-Erk pathway, and the fact that a significant fraction of T-ALL cases display constitutive MEK-Erk pathway activation, it is possible that the MEKis have a negative effect on some of the T-ALL cells themselves, especially those cases with the highest levels of MEK-Erk pathway activation. This could explain why not all T-ALL samples co-cultured in stroma benefited from MEKi treatment, since the positive effect that the MEKis had via stromal cells may have been counterbalanced by a direct negative effect on T-ALL cells. Thus, the authors could: a) evaluate whether sensitivity to the MEKis correlated with MEK-Erk activation in T-ALL cells; and b) test the effect of MEKis on T-ALL cells directly (no co-culture).

The comment of the reviewer is challenging and we could not correlate MEK/Erk activation and proliferating activity of the different T-ALL we used because we did not systematically measure MEK/ERK activation in every patient when we found out the IL18 related effect. However some other results do not clearly support a combined negative+positive effect of MEKi on T-ALL growth. The first line of arguments comes from **Figure 1-reviewer 3 + Figure 2B** (Figures removed as requested by authors) showing results from 4 individual experiments where conditioned medium from MEKi-treated MS5 cells, that does not contain active MEKi anymore due to its short half-life in vitro, does not reproducibly promote more growth than MEKi direct addition into co-cultures. As this argument might appear indirect, we also performed cultures of T-ALL with MEKi +/- MS5 cells (**Figure 2-reviewer 3**) (Figure removed as requested by authors). This experiment was done with M18 T-ALL, which is sensitive to MEKi growth stimulation and is phosphorylated on ERKs. The results show that without stromal support, T-ALL does not grow, although more cell death/apoptosis is detected compared to cultures with MS5 cells. Importantly, MEKi did not significantly modify neither cell death/apoptosis nor proliferation (shown by cell cycle analysis).

An alternative possibility is that there are T-ALL cases with low/absent IL18R surface expression, in which case these samples should be insensitive to the MEKis. IL18R levels should be correlated with sensitivity to MEKi in the co-cultures. Likewise, responsiveness to rhIL18 should be correlated with IL18R surface expression in T-ALL cells.

IL18R chain alpha and chain beta are expressed in the different T-ALL we tested, irrespective of their responsiveness to IL18 (**Figure 3-reviewer 3**) (Figure removed as requested by authors). We show to the reviewer the MFI measured by FACS of every IL18R chain. There are some differences between responsive (n=6-11) and non-responsive (n=3) samples but we are not sure this can fully explain the differences observed between both groups. Further work on the response (in terms of NFkB activation or IFNg expression) should tell us whether it is related to the strength of the signal and then we will be able to deeply investigate the mechanisms.

6. *In Figure 3B the authors show that exogenous IL18 leads to NF-kappaB activation in T-ALL cells. Does the same happen in MEKi-treated T-ALL cells in co-culture with stroma? Most importantly, is NF-kappaB activation relevant for IL18-mediated effects, i.e. if the authors pre-treat T-ALL cells with an NF-kappa B inhibitor and then co-culture them with stromal cells alone or in*

the presence of IL18, is there a blockade in the effect of the cytokine? Addressing this question would strengthen significantly the relevance of the link between IL18 and NF-kappa B.

It is for sure a very important and interesting point that represents our current projects. We have measured NFkB activation in the M105 T-ALL in different culture conditions (**Figure 4-reviewer 3**) (Figure removed as requested by authors). We have tested NFkB activation in presence of CM from MS5 cells pre-treated with or without MEKi (lanes 1 and 2). We have also measured NFkB during co-cultures with MS5/shIL18 or /shCTL cells in presence of CM from the same stromal cells pretreated or not with MEKi (lanes 3 to 8). These results show that MEKi indirectly activates NFkB in M105 T-ALL. This effect is at least partly mediated by IL18 as CM from MEKi treated MS5/shCTL cells partially rescues the drop of NFkB activation of M105 cells cultured with MS5/shIL18 stromal cells. These results are encouraging but we think there is still work to be done before we can show a solid relation between MEKi, IL18 and NFkB. We have included the CM/without MS5 EMSA part in **Figure S5** to support the link between MEKi, IL18 and NFkB and we are open to discuss with the reviewer concerning the MS5/shIL18 results.

Unfortunately we have not started yet to treat human T-ALL co-cultures with IL18 or MEKi +/- NFkB inhibitors as in our opinion this is a whole field of research that requires to be tested on numerous samples. Nevertheless we present data of one experiment in which mouse ICN1 T-ALL was cultured with NFkB inhibitors that show growth inhibition (**Figure 5-reviewer 3**) (Figure removed as requested by authors). However we do not know whether such NFkB inhibition effect is related to IL18 pathway and also if it is specific to this peculiar mouse T-ALL and can be extended to other mouse and also human T-ALL. Our future experiments will definitively be devoted to look at the relation between MEKi, IL18 and NFkB.

7. In Figure 5B, childhood T-ALL cases are compared with a mix of healthy pediatric and adult controls regarding IL18 plasma levels. To avoid the possible criticisms that there may be age-related differences in IL18 expression in healthy controls that could create a bias in the analysis, the authors should remove the adult samples and compare strictly the same age groups.

We have done this and now show the modified results in Figure 5C.

8. The actual clinical relevance of the data on the prognostic value of IL18 levels is questionable. There is a trend for better DFS in IL18 low patients (fig. 5D), but this trend is far from significant. Unless the authors provide further evidence from another, larger cohort of patients that allows for the actual identification of statistically significant differences, the data will remain speculative. Likewise, the authors should avoid using the analysis presented in panel 5E to speculate, "Also, the low IL18 group seemed to have a better outcome than the high IL18 group, whatever the EGIL phenotype (Figure 5E-F)." (page 9). The data presented in panel 5E do not add to fig.5D in terms of IL18 prognostic value per se and therefore do not allow to formally extract any conclusions on IL18 prognostic value.

Also, I have strong doubts as to whether the claim that "Combined with the EGIL stage, high IL18 plasma levels identified a patients' subset with poor outcome" (abstract). This statement does not appear to be corroborated by the data in fig. 5. If anything, the data in panel 5E identify a patient subset with low plasma levels and EGIL TIII stage that present good prognosis. However, this is also arguable, because the authors compare EGIL TIII versus not TIII within the IL18 low-expressing cases. In such way, the prognostic variable is the EGIL stage (measured within a defined T-ALL subgroup with low IL18 levels), not IL18. If the authors are to claim that the combination of the two discrete variables (IL18 levels and EGIL stage) is of prognostic value, it seems to me that they must compare the four groups in a single statistical analysis (i.e. Hi IL18-EGIL TIII x Hi IL18-EGIL non TIII x Lo IL18-EGIL TIII x Lo IL18-EGIL non TIII). Otherwise, the clinical importance of their analyses will be limited.

We analyzed back our data and performed measurements on additional plasma recovered from supplementary patients. The number is now 92 patients and 29 controls, all being children. The analysis still shows significant differences between these two cohorts (see Figure 5C). Moreover, and as commented by the reviewer, we have performed comparative analysis of the 4 groups (total 81 patients with accessible clinical data) using Cox regression, following advices by a statistician. This analysis uncovers a group of patients being EGIL TIII and IL18low with a better prognosis,

although the analysis is not significant compared to the other groups, except with the EGIL not TIII, IL18low. We have now modified the abstract sentence and also modified Figure 5 and text according to reviewer's advices and hope that it fits with the reviewer criticisms.

Minor

1. *There are typos in the EMSA section of the Methods.*

The typos have been corrected.

2. *Figure 5 legend, panel D: "(...) and low (IL18 levels above the median(...)). "above" should be corrected to "below" or "under". Also, in page 9, last sentence, the reference to figure 5E-F is incorrect. There is no panel F in this figure.*

This has been corrected.

3. *Figure 3C is somewhat "overcrowded" and difficult to follow. Similar to other panels, an inset with the data on day 28 would help making the results easier to read.*

The figure has been modified accordingly to reviewer 3 comments.

4. *Statistics for Figure S6 are missing.*

Statistics are now in Figure S6.

3rd Editorial Decision

13 February 2014

Thank you for the submission of your revised manuscript to EMBO Molecular Medicine.

We have now received the enclosed reports from the Reviewers that were asked to re-assess it. As you will see that while the reviewers are now supportive a few issues remain that require your action.

Reviewer 2 would like you to clarify the details of new data in Fig 5B as suggested.

Reviewer 3, instead, would like you to altogether remove the data illustrated in Fig.5E. I do appreciate his/her points but I would rather suggest a compromise in that you be more critical of these data by commenting on their limitations, and move them into the supplemental information.

If you comply with the requested changes, I am prepared to make a final decision at the Editorial level. To also ensure that your next revised version is the final one, I am also asking you at this time to address the following pending additional items:

1) We would need a short list (up to 5) of bullet points that summarize the key NEW findings. The bullet points should be designed to be complementary to the abstract and will be used online in our new online platform.

2) As per our Author Guidelines, the description of all reported data that includes statistical testing must state the name of the statistical test used to generate error bars and P values, the number (n) of independent experiments underlying each data point (not replicate measures of one sample), and the actual P value for each test (not merely 'significant' or 'P < 0.05').

3) We note that a) particulars of Fig. 3F are cut-off and b) the upper panel in Fig. 5B is a bit blocky. Please provide corrected/improved figures.

4) We are now encouraging the publication of source data, particularly for electrophoretic gels and blots, with the aim of making primary data more accessible and transparent to the reader. Would you be willing to provide a PDF file per figure that contains the original, uncropped and unprocessed scans of all or at least the key gels used in the manuscript? The PDF files should be labeled with the appropriate figure/panel number, and should have molecular weight markers; further annotation may

be useful but is not essential. The PDF files will be published online with the article as supplementary "Source Data" files. If you have any questions regarding this just contact me.

Please take extra care in ensuring that all figure callouts in the manuscript text and the supplementary information TOC are appropriately amended and also please provide an additional copy of the revised manuscript with all changes highlighted. This will ensure that your manuscript is evaluated as quickly as possible.

I look forward to reading a new revised version of your manuscript as soon as possible.

***** Reviewer's comments *****

Referee #1 (Remarks):

Is suitable for publication

Referee #2 (Comments on Novelty/Model System):

The work has been conducted using what are currently state of the art in vitro and in vivo models systems for study of primary human T-ALLs. There are no ethical issues.

Referee #2 (Remarks):

The authors have tried to be responsive to the initial critique. The use of diverse samples/tumors in various experiments is less than ideal, but the limitations that are faced when using primary samples are real, and some leeway can be granted here. The information in Table 1 has been clarified. Most importantly, the authors now provide data in Figure 5B that IL-18 production can be detected in the marrow cells of NSG mice. The details of this new panel require clarification. It is not clear from the text or the figure legend if these experiments were performed with tumor bearing or tumor free mice; both might be interesting, as a point of comparison. Others comment have been adequately addressed.

Referee #3 (Remarks):

Uzan et al have made a serious effort to adequately reply to the previous criticisms. The most important points have been properly addressed.

My reservations remain in what concerns the relevance of the data on the prognostic value of IL18 plasma levels in face of the EGIL stage. The new statistical analysis the authors have now performed reinforces in fact my initial impression that the IL18 low Stage III patients with "good prognosis" can only be defined as such by comparison strictly with IL18 low patients not stage III, whereas if IL18 levels in combination with EGIL stage were to identify any subgroup with clear prognostic relevance it should be identified as such by comparison also with all the remaining subgroups (i.e. the two High IL18 subgroups (TIII and not TIII) in this case). Given the overall quality, originality and high relevance of the studies by Uzan et al, and most importantly their clinical potential in identifying IL-18 produced by the microenvironment as a novel targetable axis involved in T-ALL disease development, I believe that the prognosis data shown in Figure 5E does not provide an added value to the manuscript. On the contrary, I fear it may be misleading and thus weaken the paper. I would suggest removing these data.

Please find enclosed the final revised version of our manuscript entitled “**Interleukin-18 produced by bone marrow derived stromal cells supports T cell acute leukemia propagation**” by B Uzan *et al.* for publication as a regular article in *EMBO Molecular Medicine*. It still comprises 5 composite figures and 2 Tables.

The supplementary files have been extended to 9 figures and 2 supplementary Tables.

Compared to the previous version, the main changes can be visualized in Figure 5 in which we took away the Figure 5E, as suggested by the reviewer 3, and included it as a supplementary figure 9, as you suggested. We have answered reviewer 2 comments on IL18 production by mouse BM cells with or without leukemia infiltration directly in the text. As we did not perform many measurements of IL18 produced by mouse BM cells in both conditions we are not comfortable enough to indicate differences. This shall definitively be tested in coming experiments.

In terms of writing, we took off some parts (in the end of the abstract) mainly concerning the comments on the prognosis impact of IL18 levels in patients or we modified to tune down the importance of these findings at the end of the results section, and also in the discussion. Where appropriate sentences are underlined to show you the changes. Also we looked closely to the number of experiments in figure legends and we changed the “stars” to incorporate the value of “p” for statistical significance of experiments directly in the figures. We now include a copy of the raw data of western blots we show in the figures.

Also, as required, we include a “bullet points” section after the abstract.

I hope this version will be finally accepted for publication in *EMBO Molecular Medicine*.